# Correcting glucose-6-phosphate dehydrogenase deficiency with a small-molecule activator

Sunhee Hwang [1], Karen Mruk [1,2,6], Simin Rahighi[3,7], Andrew G. Raub[1,4], Che-Hong Chen[1], Lisa E. Dorn [1,8], Naoki Horikoshi[3], Soichi Wakatsuki[3,5], James K. Chen [1,2] & Daria Mochly-Rosen [1]

Glucose-6-phosphate dehydrogenase (G6PD) deficiency, one of the most common human genetic enzymopathies, is caused by over 160 different point mutations and contributes to the severity of many acute and chronic diseases associated with oxidative stress, including hemolytic anemia and bilirubin-induced neurological damage particularly in newborns. As no medications are available to treat G6PD deficiency, here we seek to identify a small molecule that corrects it. Crystallographic study and mutagenesis analysis identify the structural and functional defect of one common mutant (Canton, R459L). Using high-throughput screening, we subsequently identify AG1, a small molecule that increases the activity of the wild-type, the Canton mutant and several other common G6PD mutants. AG1 reduces oxidative stress in cells and zebrafish. Furthermore, AG1 decreases chloroquine- or diamide-induced oxidative stress in human erythrocytes. Our study suggests that a pharmacological agent, of which AG1 may be a lead, will likely alleviate the challenges associated with G6PD deficiency.

[1] Department of Chemical and Systems Biology, Stanford University School of Medicine, Stanford, CA 94305, USA. [2] Department of Developmental Biology, Stanford University School of Medicine, Stanford, CA 94305, USA. [3] Department of Structural Biology, Stanford University School of Medicine, Stanford, CA 94305, USA. [4] Department of Chemistry, Stanford University, Stanford, CA 94305-5080, USA. [5] Photon Science, SLAC National Accelerator Laboratory, Menlo Park, CA 94025-7015, USA. [6] Present address: University of Wyoming School of Pharmacy, 1000 E. University Ave., HS 596, Laramie, WY 82071, USA. [7] Present address: Chapman University School of Pharmacy (CUSP), Harry and Diane Rinker Health Science Campus, Chapman University, Irvine, CA 92618, USA. [8] Present address: The Ohio State University College of Medicine, 473 W 12th Ave, Columbus, OH 43210, USA. Correspondence and requests for materials should be addressed to D.M.-R. (email: mochly@stanford.edu)

Reduced glutathione (GSH) provides the cellular first line of defense against oxidative stress-induced injury, which can be maintained by NADPH generated mainly via the pentose phosphate pathway and its rate-limiting enzyme, glucose-6-phosphate dehydrogenase (G6PD; Fig. 1a). Accordingly, missense DNA mutations that impair G6PD activity or stability result in increased oxidative stress and a spectrum of disease phenotypes featuring, most commonly, hemolytic anemia, and collectively called G6PD deficiency[1]. In particular, G6PD is essential in preserving the integrity of erythrocytes because, lacking mitochondria, they have no other NADPH-generating enzymes to protect against oxidative stress[2].

G6PD deficiency represents one of the most common inherited and sex-linked enzymopathies. The G6PD gene maps to the X-chromosome; thus, the phenotype is manifest fully in males whereas female heterozygotes display varying degrees of G6PD deficiency, due to alternate X-chromosome inactivation[3,4]. G6PD deficiency afflicts more than an estimated 400 million individuals worldwide, many of whom live in malaria endemic regions. Impaired anti-oxidant defense in G6PD-deficient erythrocytes makes them vulnerable to early membrane damage and ultimately phagocytosis when infected with malaria[5]. This mechanism that results in limiting the propagation of the parasite in the bloodstream explains how G6PD deficiency provides resistance against malaria[4–6].

Symptoms of G6PD deficiency are triggered by exposure to certain foods, medications, infection and/or environmental factors[2,4]. For example, consumption of fava beans causes oxidative stress to erythrocytes possibly by two main toxins, vicine or convicine, thus triggering acute hemolytic episode (favism) in affected subjects[7]. G6PD-deficient individuals are also at a high risk of severe hemolysis when given anti-malarial drugs, such as quinine, primaquine or chloroquine, through irreversible oxidative activity of their metabolites on erythrocytes[8,9]. G6PD deficiency can be life-threatening, especially in newborns, leading to bilirubin-induced neurological injury and bilirubin encephalopathy (kernicterus) and even to death[10–15]. In a recent study, a systematic analysis of 2253 articles discussing G6PD revealed that dysregulation of G6PD is also associated with autoimmune diseases and metabolic disorders, indicating that clinical risks associated with G6PD deficiency are likely underestimated[16]. As there are currently no means to correct G6PD deficiency, there is no treatment for the disease; management mainly consists of supportive care and discontinuation of triggers. The use of anti-oxidants such as vitamin E or selenium has proven to be ineffective in treating G6PD deficiency[17–19].

On the other hand, based on studies performed in Sardinia, G6PD deficiency has some beneficial effect on longevity[20]. In studies using dehydroepiandrosterone (DHEA), an inhibitor of G6PD (that is also an adrenal steroid), loss of G6PD function has been suggested to prevent cancer progression[20,21]. Despite all the beneficial effects, the detrimental effect of G6PD deficiency are still clear in hemolytic anemia and kernicterus of infancy. Given the risk of hemolytic crisis and related sequelae from various triggers in affected subjects in both malaria endemic and non-endemic regions, we believe that developing a pharmacological agent that corrects G6PD deficiency may benefit affected individuals.

G6PD is functionally active as a dimer or a tetramer[22]. Each monomer has a catalytic nicotinamide adenine dinucleotide phosphate (NADP$^+$)-binding domain and β+α domain, containing an additional binding site for NADP$^+$ that structurally stabilizes the enzyme (Fig. 1b)[23,24]. The glucose 6-phosphate (G6P)-binding site is located between these two domains (Fig. 1b). Using different informatics tools, we recently demonstrated that the majority of the variants that cause severe (<10%

of normal G6PD activity, class I and class II) or mild (10–60% of normal G6PD activity, class III) deficiency are primarily located in those functional regions of the enzyme, disturbing the enzyme's activity and stability[25].

Here we present a comprehensive report describing the molecular basis for G6PD deficiency with Canton variant (R459L) and our efforts to restore its decreased function using a small molecule, AG1. Our study provides the first step in identifying a potential therapeutic approach to correct G6PD deficiency.

## Results

**Canton G6PD variant has reduced activity and stability.** We first began our efforts focusing on the Canton variant, with the mutation R459L, located in the β+α domain (Fig. 1b). Canton G6PD is prevalent in China and Southeast Asia (50–60% of the variants[26,27]), causing severe deficiency (class II). Recombinant Canton G6PD enzyme showed only 18% of normal G6PD activity with lower $K_M$ for both NADP$^+$ and G6P (Fig. 1c), which is consistent with the biochemical characteristics analyzed using blood samples derived from subjects carrying the Canton variant[28]. Relative to the wild-type (WT) enzyme, Canton G6PD also displayed impaired ability to form tetramers in the presence of increased concentrations of the cofactors that facilitate tetramer formation, NADP$^+$ or MgCl$_2$[29,30], when cross-linked by glutaraldehyde (Supplementary Fig. 1a). This reduced oligomerization state of the Canton G6PD may contribute to its reduced enzymatic activity. Furthermore, the Canton variant was less thermostable; its $T_{1/2}$, the temperature at which the enzyme retains half of its catalytic activity, was 42.6 °C, which is 4.6 °C lower than $T_{1/2}$ of WT enzyme (Fig. 1d). Canton G6PD was more susceptible to degradation by chymotrypsin relative to the WT enzyme, which corresponded with a significantly decreased enzymatic activity (Fig. 1e). This suggests that Canton G6PD may undergo higher conformational fluctuation, leading to a greater accessibility to proteases and its reduced thermostability[31].

We also show that the Canton variant was less stable than WT G6PD in lymphocytes derived from a male subject carrier with a Canton mutation in G6PD; 24 h after cycloheximide treatment (50 μg mL$^{-1}$) to inhibit de novo protein biosynthesis, the level of the Canton variant protein was ~33% lower than the WT enzyme (Fig. 1f and Supplementary Fig. 1b). G6PD activity in lysates of lymphocytes of the Canton variant carrier was ~90% lower than G6PD activity in normal lymphocytes (Fig. 1g), which coincided with low levels of total GSH and increased levels of reactive oxygen species (ROS) (Fig. 1h, i). Moreover, the viability of the lymphocytes of the Canton variant carrier was ~50% lower relative to normal lymphocytes when cultured under the stress condition induced by serum starvation (Fig. 1j).

The same results were observed in SH-SY5Y neuronal cells transiently expressing WT G6PD or the Canton variant (His-tagged); Canton G6PD protein levels dropped by 50% within 24 h of cycloheximide treatment, as compared to 20% decrease of WT G6PD (Supplementary Fig. 1c). SH-SY5Y cells expressing the Canton variant also showed lower G6PD enzymatic activity in the lysates, and a lower level of GSH, a higher level of ROS, and lower cell viability in culture (Supplementary Fig. 1d, e, f, g) and knockdown of G6PD by siRNA reduced cell viability, recapitulating the phenotype of G6PD deficiency (Supplementary Fig. 1h).

**Canton mutation loses essential inter-helical interactions.** To elucidate the molecular basis of the reduced stability and activity of Canton G6PD, we examined the crystal structures of WT and Canton G6PD at 1.9 Å and 2.6 Å resolution, respectively (Table 1). Although no NADP$^+$ was added to protein solution prior to the crystallization, both crystal structures contained

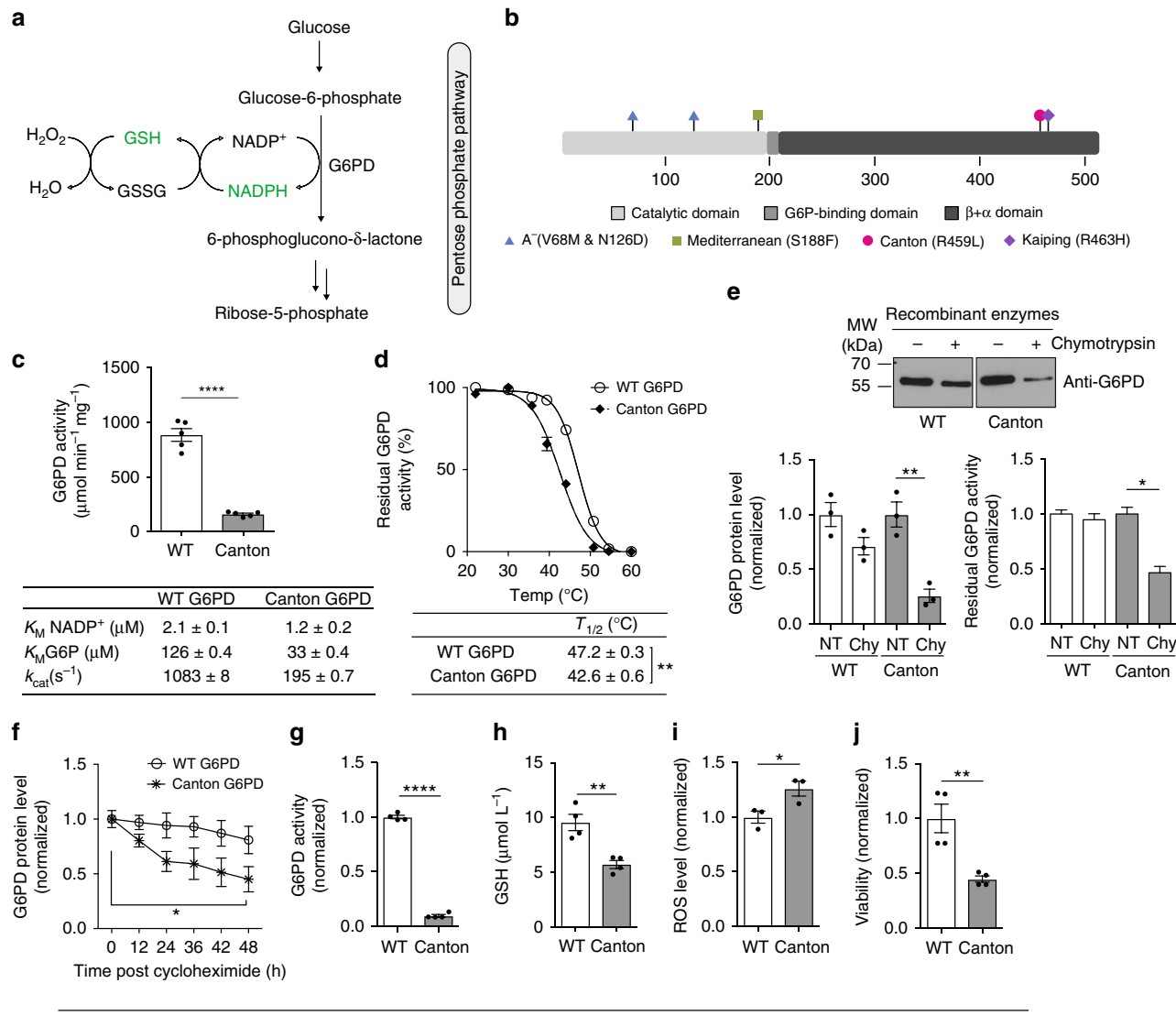

**Fig. 1** Canton G6PD (R459L) variant is biochemically different from WT G6PD. **a** Enzymatic scheme of G6PD activity. **b** A linear map of G6PD domain structure with most common variants indicated. **c** Catalytic activity of recombinant WT G6PD and Canton G6PD enzymes with kinetic parameters ($n = 5$, ****$p < 0.0001$). **d** Thermostability of WT G6PD and Canton G6PD enzyme ($n = 3$, **$p = 0.003$). **e** G6PD protein levels and residual G6PD activity (normalized to NT (no treatment) of each enzyme) after incubation with chymotrypsin for 1 h ($n = 3$ for protein level assay, **$p = 0.0046$; $n = 2$ for enzyme assay, *$p = 0.024$). **f** Protein stability assessment with cycloheximide treatment ($50 \mu g \, mL^{-1}$), blocking de novo protein biosynthesis, in lymphocytes derived from corresponding subjects ($n = 3$, *$p = 0.013$). Protein levels were normalized to the level of each enzyme at 0 h (no treatment). **g** G6PD activity was lower in cell lysates with Canton variant ($n = 4$, ****$p < 0.0001$). **h, i, j** Lymphocytes with Canton variant generated less GSH and more reactive oxygen species (ROS) and were less viable ($n = 4$, ($n = 3$ for Fig. 1i), *$p < 0.05$, **$p < 0.01$). Error bars represent mean ± SEM. Statistical differences were calculated by two-tailed unpaired Student's $t$-test. NT: no treatment, WT: wild-type, Chy: chymotrypsin

NADP$^+$ at the second NADP$^+$-binding site (structural NADP$^+$-binding site) in the β+α domain (Supplementary Fig. 2a). (The stability of the enzyme with NADP$^+$ association at this site has been previously reported and NADP$^+$ can be removed by incubating the enzyme with G6P[24,30].) The overall conformations of WT and Canton G6PD were very similar, as indicated by a root-mean-square deviation of 0.6 Å for the superimposition of Cα atoms (Fig. 2a). However, we noticed a loose helical interaction between the helix (αn), containing the Canton variant-R459L, and an adjacent helix (αe) (Fig. 2b, left panel), which was not described in previously reported structures[24]. In WT G6PD, R459 forms electrostatic and hydrogen bond interactions with D181 and N185 on the adjacent helix (αe), whereas Canton mutation does not, resulting in loosened inter-helical interactions and displacements of the helix (αe) and a proceeding loop consisting

of K171, P172, F173, G174, and R175, several amino acids away from R459-interacting residues on the helix (αe) (Fig. 2b, right panel). In particular, K171 and P172 are the key residues involved in positioning of G6P and NADP$^+$ in their binding pockets[23,24]. Thus, the loose inter-helical interaction in the Canton variant is likely to be a major driving force for positioning the loop and thus changing the orientations of these residues. In 7 out of 8 molecules in the asymmetric unit of the Canton variant structure, P172 was observed in the *trans* conformation, consistent with previous data[23,24] and accordingly, the side chain of K171 was oriented away from catalytic NADP$^+$ and G6P binding pockets (Fig. 2b, right panel, and Supplementary Fig. 2b, c). K171A and P172G mutations in WT G6PD completely abolished the enzyme's catalytic activity, indicating the importance of these residues in catalysis (Supplementary Fig. 2d).

**Table 1 Diffraction data and refinement statistics**

|  | G6PD WT | G6PD Canton (R459L) |
|---|---|---|
| *Data collection* |  |  |
| Beamline | SSRL BL12-2 | ALS BL 5.0.2 |
| Space group | *F*222 | *P*2₁2₁2₁ |
| Cell dimensions |  |  |
| *a, b, c* (Å) | 60.6, 173.4, 215.9 | 127.1, 206.2, 211.8 |
| *α, β, γ* (°) | 90.0, 90.0, 90.0 | 90.0, 90.0, 90.0 |
| Wavelength (Å) | 1.0 | 1.0 |
| Resolution (Å) | 50.0–1.9 (1.95–1.90) | 50.0–2.6 (2.74–2.60) |
| CC (½) | 1.00 (0.82) | 1.00 (0.52) |
| $R_{merge}$ | 0.12 (0.79) | 0.15 (1.44) |
| $I/\sigma I$ | 11.0 (3.0) | 10.5 (1.6) |
| Completeness (%) | 95.3 (91.3) | 100.0 (100.0) |
| Redundancy | 5.7 (5.7) | 7.3 (7.5) |
| *Refinement* |  |  |
| Resolution (Å) | 50.0–1.9 | 50.0–2.6 |
| No. reflections | 37,018 | 171,050 |
| $R_{work}/R_{free}$ | 17.2/21.2 | 18.7/21.1 |
| No. atoms |  |  |
| Protein | 3946 | 31,554 |
| Water | 406 | 537 |
| Other | 78 | 598 |
| *B*-factors (average) |  |  |
| Protein | 24.6 | 36.3 |
| Water | 34.9 | 45.7 |
| Other | 38.8 | 57.7 |
| R.m.s. deviation |  |  |
| Bond lengths (Å) | 0.015 | 0.014 |
| Bond angles (°) | 1.786 | 1.734 |
| PDB ID | 6E08 | 6E07 |

We further mutated the R459-interacting residues, D181 and N185, to alanine in WT G6PD to determine the importance of the inter-helical interactions and found that these point-mutated enzymes are biochemically similar to the Canton variant; they exhibited about 20% of the normal activity, with lower $K_M$ values for both NADP⁺ and G6P and lower $T_{1/2}$ values (Fig. 2c and Supplementary Fig. 2e). These mutants were also more susceptible to proteolytic digestion, relative to WT G6PD (compare Fig. 2d to Fig. 1e). Taken together, these data support the importance of the inter-helical interaction between R459 and D181 and N185 for catalytic activity and stability of the enzyme and provides a structural insight into the biochemical defects of the Canton variant. Indeed, other human mutations located around this helical interaction site, such as P172S (<10% enzyme activity), F173L (<10% enzyme activity), and D181V (10–60% enzyme activity), cause moderate to severe G6PD deficiency[32–34], likely because they are predicted to undergo similar conformational changes based on our observation.

**AG1 activates and stabilizes G6PD mutants.** Our biochemical and structural studies led us to determine whether improving the activity of G6PD variants with a pharmacological agent can provide a new therapeutic approach to reduce the risk of pathologies implicated in patients with G6PD deficiency. To this end, we screened for agonists of G6PD (AGs) using the recombinant Canton G6PD enzyme by a high-throughput screen (HTS) and identified one agonist, 2,2'-disulfanediylbis(*N*-(2-(1*H*-indol-3-yl)ethyl)ethan-1-amine) (AG1, $M_r = 438.1912$) (Supplementary Fig. 3a). We confirmed that the active species in the HTS sample is a product resulting from thiol oxidation under ambient conditions[35], which was validated by mass spectrometry and nuclear magnetic resonance spectroscopy (Supplementary Fig. 3a and Supplementary Note 1). AG1 increases the activity of Canton

G6PD up to 1.7-fold with EC₅₀≈3 μM and WT G6PD by about 20% over basal activity (Fig. 3a, b and Supplementary Fig. 3b). Although AG1 was a mild activator, it changed the kinetic parameters of Canton G6PD, indicating that AG1 may facilitate improved binding of NADP⁺ and/or G6P to the enzyme (Fig. 3c). We found that AG1 also promoted formation of dimers, as determined by partial native gel electrophoresis (Fig. 3d). The small increase observed in molecular weight of the monomeric G6PD might be due to either some modification of the enzyme by AG1 or an equilibrium shift toward dimeric states. When the recombinant Canton G6PD enzyme was incubated with G6P to remove the structural NADP⁺, the dimeric G6PD was destabilized, dissociating into a monomer (Fig. 3e). In the presence of AG1, however, the equilibrium shifted toward a dimer (Fig. 3e), further indicating that AG1 stabilizes a dimeric form of the enzyme. AG1 had no effect on the dimerization or activity of several other NAD- or NADP⁺-dependent dimeric or tetrameric enzymes, including 6-phosphogluconate dehydrogenase (6PGD), glyceraldehyde 3-phosphate dehydrogenase (GAPDH), aldehyde dehydrogenase 2 (ALDH2), and aldehyde dehydrogenase 3A1 (ALDH3A1) (Supplementary Fig. 3c). Whereas 1 μM of AG1 increased the viability of SH-SY5Y cells by 20%, it had no effect when G6PD was knocked down by siRNA, supporting the specificity of AG1 toward G6PD (Supplementary Fig. 3d). Note also that the decrease in viability by the knockdown of G6PD implies the critical role of G6PD for cell survival. AG1 also reduced the susceptibility of Canton G6PD to proteolysis (Fig. 3f) and mildly improved its stability in lymphocytes (Fig. 3g and Supplementary Fig. 3e). The enzymatic activity in lymphocyte lysates with the Canton variant (~10% of normal activity) was enhanced by 78% in the presence of AG1 (Fig. 3h). The levels of GSH were slightly higher after treatment with AG1 for 24 h, which coincided with decreased ROS levels and improved cell viability (Fig. 3i–k) in the lymphocytes cultured under the serum starvation stress. AG1 treatment also increased the proteolytic stability of Canton G6PD in SH-SY5Y cells (Supplementary Fig. 3f). There was also a mild increase in G6PD activity, GSH levels, and viability of these cells together with a decrease in ROS levels (Supplementary Fig. 3g, h, i, j). Finally, as predicted, the activity of Canton variant-mimicking mutations that we generated (Fig. 2c, d), D181A and N185A, were similarly affected by AG1, as measured by increased activity and proteolytic stability (Supplementary Fig. 4a, b, c), suggesting that AG1 may correct a structural defect common to mutations that disturb that helical structure (Fig. 2b).

Because G6PD exhibits cooperative folding[36] and AG1 increased and/or stabilized G6PD dimer levels, we next examined the possibility that binding of AG1 to the enzyme could correct other point mutation-containing G6PD variants outside the αe-αn inter-helical interaction sites. We focused on A⁻ (V68M & N126D), Mediterranean (S188F), and Kaiping (R463H) G6PD (Fig. 1b), the three most common human variants in non-overlapping regions of the world causing mild to severe deficiency (Africa, Mediterranean and Southeast Asia, respectively). AG1 activated all these variants by up to 2-fold and increased the levels of the dimeric state (Fig. 3l, m). AG1 also stabilized G6PD in fibroblasts derived from a subject who carries Mediterranean mutation following cycloheximide treatment (Fig. 3n and Supplementary Fig. 5a). The lysate of human fibroblasts with the Mediterranean variant had only 22% activity of control human fibroblasts, but AG1 increased that activity by 50% (Supplementary Fig. 5b). Fibroblasts carrying the Mediterranean variant generated more ROS and less GSH under stress induced by serum starvation relative to fibroblasts from control subject, which was significantly blunted by AG1 treatment (Fig. 3o, p). Lower cell viability under the same condition was also improved by 22% in the presence of AG1 (Supplementary Fig. 5c). When Mediterranean G6PD expression was knocked down by siRNA, the viability was

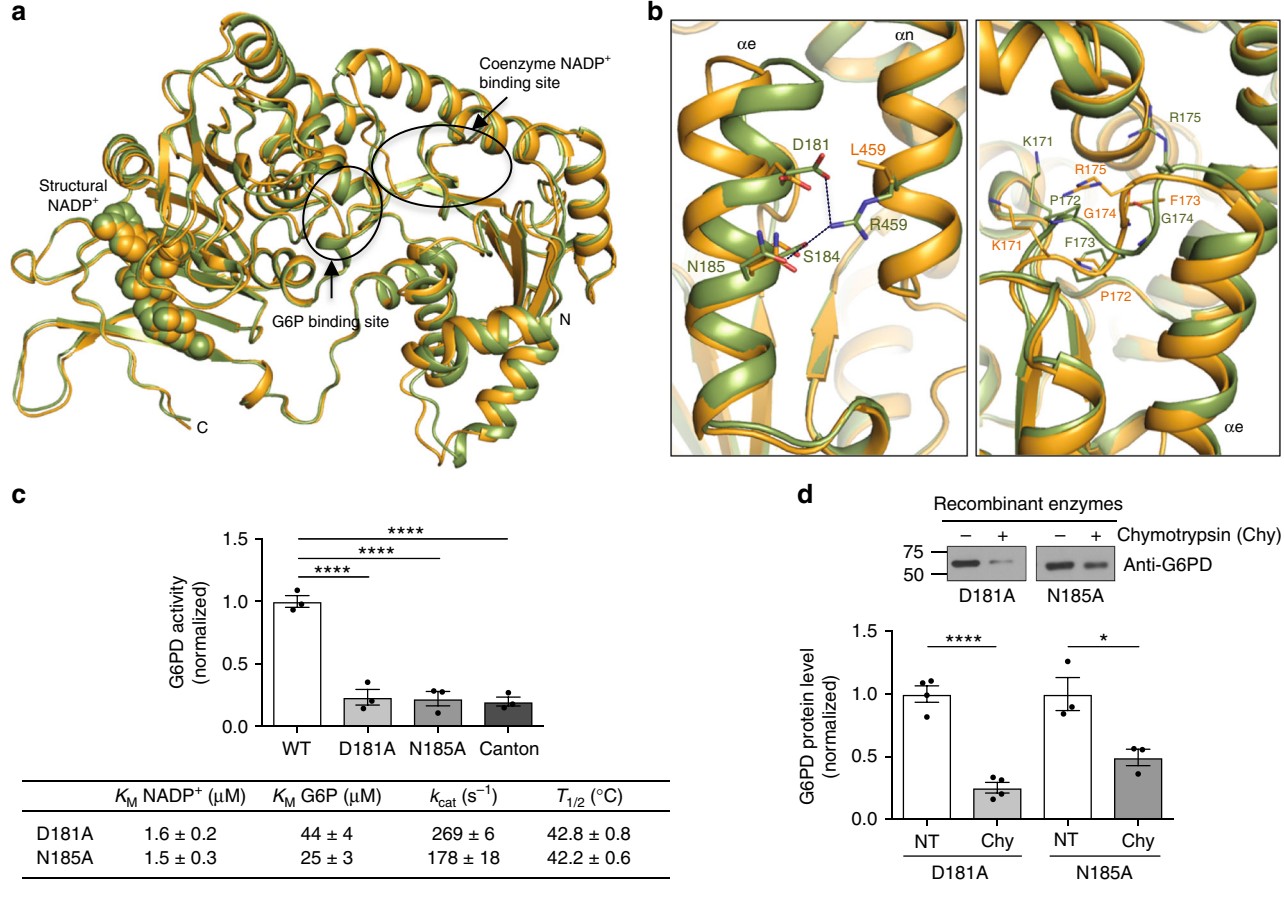

**Fig. 2** Canton mutation (R459L) loses essential inter–helical interactions. **a** Structural overlay of WT G6PD (green) and Canton variant (orange). Structural NADP$^+$ is shown as spheres, and arrows and circles indicate G6P and catalytic NADP$^+$-binding sites (G6P and catalytic NADP$^+$ were not observed in our structures). **b** (Left) Inter-helical interactions through R459 on the αn helix in WT G6PD and side chains of D181 and N185 on the adjacent helix (αe). (Right) Canton mutation loses such interactions, leading to displacements of the helix (αe) and a loop containing K171, P172, F173, G174 and R175 that precedes the helix. **c**, **d** Mutations of R459-interacting residues on the αe helix showed Canton mutation-like activity and thermostability ($n = 3$, $^{****}p < 0.0001$, one-way ANOVA) and were also susceptible to chymotrypsin treatment ($n = 4$ for D181A, $^{****}p < 0.0001$; $n = 3$ for N185A, $^{*}p = 0.026$, two-tailed unpaired Student's $t$-test). Error bars represent mean ± SEM. NT no treatment; WT: wild-type; Chy: chymotrypsin

further dropped by 23%, and AG1 did not rescue it (Supplementary Fig. 5c), indicating the selective effect of AG1 for G6PD. Taken together, these data raise the possibility that by increasing (or stabilizing) the levels of dimeric G6PD and/or increasing the enzyme activity, AG1 may represent a lead compound for a drug to treat not only Canton-like mutations, but also some of the other most common G6PD deficiencies in humans.

**AG1 reduces oxidative stress in zebrafish**. Capitalizing on a recent report describing a morpholino-generated G6PD-deficient zebrafish model, in which exposure to pro-oxidants resulted in cardiac edema and a brisk hemolysis[37], we next set out to determine the effect of AG1 in vivo. Using zebrafish embryos, we first confirmed that embryos develop normally at concentrations <10 μM of AG1 (Fig. 4a and Supplementary Fig. 6a), indicating that AG1 is not toxic to developing zebrafish embryos. We next used the anti-malarial drug, chloroquine, a common trigger for hemolytic crisis in G6PD-deficient humans[4,37], to induce an oxidative challenge in the zebrafish embryos and found that chloroquine (100 μg mL$^{-1}$) treatment at 24 h post fertilization (hpf) led to pericardial edema and increased ROS levels (Fig. 4a, b), which was consistent with the primaquine-induced phenotypes observed in a morpholino-based G6PD-deficient zebrafish model[37]. Upon chloroquine treatment, hemoglobin staining was also slightly

reduced (Supplementary Fig. 6b). Under the same condition, AG1 significantly reduced ROS levels, resulting in less embryos exhibiting pericardial edema (Fig. 4a, b and Supplementary Fig. 6a; scores were determined by an observer blinded to the treatment group) and mildly increased hemoglobin staining (Supplementary Fig. 6b). Although a slight increase in G6PD activity was observed in lysates of pooled AG1-treated embryos, there was a significant increase in total NADPH levels possibly due to the increase in the product of G6PD downstream, 6-phosphogluconate, which serves as a substrate for the downstream enzyme, 6-phosphogluconate dehydrogenase (6PGD), another NADPH-producing enzyme in the pathway (Fig. 4c). As expected, the attenuation of pericardial edema was specific to G6PD deficiency, as pericardial edema due to mesoderm defects in *tbx16* mutants was not corrected by AG1 treatment (Supplementary Fig. 6c, d)[38,39]. To further confirm the specificity of AG1 in vivo, we used CRISPR-Cas9 to generate loss-of-function F0 embryos (crispants). *g6pd* crispants had a lower G6PD level (Supplementary Fig. 6e), a 51% higher level of ROS, a 67% lower level of G6PD activity and a 58% lower NADPH level, and increased pericardial edema (Fig. 4d–f; scores were determined by an observer blinded to the treatment group). Treatment with 1 μM AG1 did not significantly affect these parameters in the *g6pd* crispants (Fig. 4d–f). Note also that there was a slight (albeit not statistically significant) increase in the number of crispant embryos exhibiting reduced hemoglobin staining (Supplementary Fig. 6f).

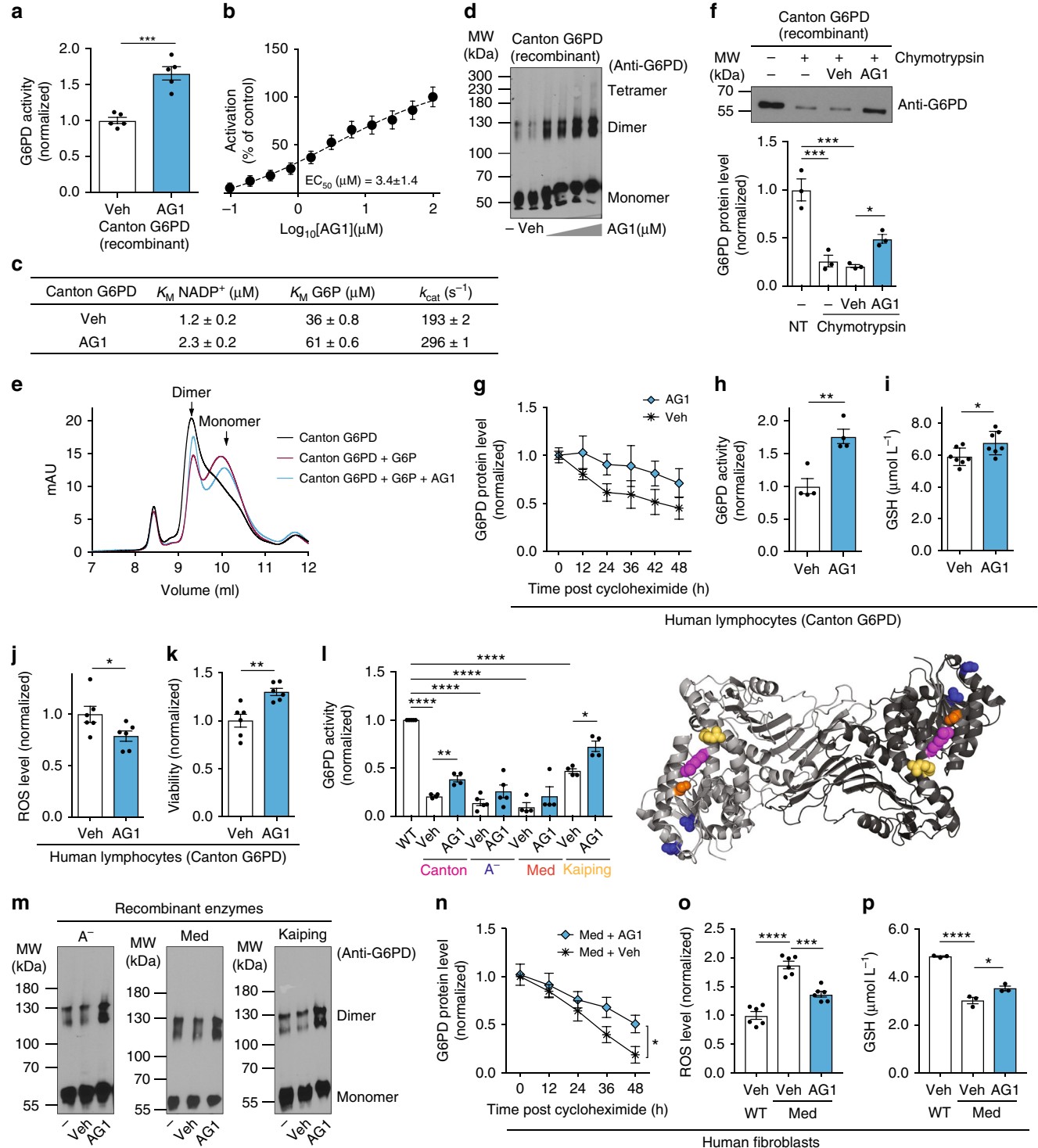

**AG1 reduces hemolysis of human erythrocytes**. We next determined whether AG1 protects erythrocytes from oxidative stress. Our preliminary study using human erythrocytes from seven healthy subjects showed that AG1 (5 μM) reduces the extent of hemolysis, when erythrocyte suspension (5%) was exposed to either 1 mM chloroquine (CQ) or diamide (a GSH oxidant), suggesting anti-hemolytic potential of AG1 (Fig. 5a). In support of this, AG1 increased GSH levels and reduced ROS levels together with increased G6PD activity under these drug-induced oxidative stress (Fig. 5b–d). Oxidative stress impairs

erythrocyte membrane integrity through initial oxidation of hemoglobin, leading to the precipitation of Heinz bodies and band 3 (a major erythrocyte membrane protein) clustering; thus, band 3 clustering serves as an essential molecular marker of erythrocyte removal[40]. Using erythrocytes isolated from 9 to 11 individual whole blood samples, we confirmed that band 3 protein is aggregated with either chloroquine (CQ) or diamide treatment, which was alleviated by AG1 treatment (Fig. 5e and Supplementary Fig. 7a). This suggests that AG1 contributes to stabilizing erythrocyte membranes. Red blood cell transfusion is

**Fig. 3** AG1 (activator of G6PD) induces biochemical changes in the Canton variant. **a** Increased activity of Canton G6PD enzyme by AG1 ($n = 5$, ***$p = 0.0002$, two-tailed unpaired Student's $t$-test) and **b** a dose response curve of AG1. **c** AG1 changed kinetic parameters of Canton G6PD. **d** AG1 promoted dimerization of Canton G6PD ($n = 3$). **e** Size-exclusion FPLC (calibrated Superdex 75 10/300 GL column) profile of purified Canton G6PD in the presence of G6P or AG1. **f** AG1 reduced proteolytic susceptibility of Canton G6PD ($n = 3$, ***$p < 0.001$, *$p < 0.05$, one-way ANOVA). The protein levels were normalized to the non-treated (NT) enzyme level. **g** Cycloheximide-chase assay using lymphocytes carrying the Canton variant ($n = 3$). Protein levels were normalized to the level of each enzyme at time 0 h. **h**, **i**, **j** AG1 increased G6PD activity in cell lysates with the Canton variant ($n = 4$, **$p = 0.0032$, two-tailed unpaired Student's $t$-test), mildly enhanced a GSH level ($n = 7$, *$p = 0.0282$, two-tailed unpaired Student's $t$-test) and reduced a ROS level in culture ($n = 6$, *$p = 0.0452$, two-tailed unpaired Student's $t$-test). **k** AG1 increased viability of lymphocytes carrying the Canton variant ($n = 6$, **$p = 0.003$, two-tailed unpaired Student's $t$-test). **l**, **m** AG1 activated other major G6PD variants, including A− (V68M, N126D; blue spheres), Mediterranean (S188F, orange spheres), and Kaiping (R463H, yellow spheres) variants, respectively ($n = 4$, ****$p < 0.0001$, **$p < 0.01$, *$p = 0.011$, one-way ANOVA), and promoted their dimerization ($n = 3$). Purple spheres in the structure represent the side chain of R459. **n** Cycloheximide-chase assay using fibroblasts carrying the Mediterranean variant ($n = 4$, *$p = 0.0437$, two-tailed unpaired Student's $t$-test). **o**, **p** AG1 significantly decreased a ROS level ($n = 6$, ***$p = 0.0001$, ****$p < 0.0001$, one-way ANOVA) and increased a GSH level in those cultures ($n = 3$, *$p = 0.0214$, ****$p < 0.0001$, one-way ANOVA). 100 μM and 1 μM of AG1 were used for in vitro assays and cell-based assays, respectively. 5% DMSO (stock) was used as vehicle. For FPLC assay, 500 μM AG1, 200 μg of Canton G6PD recombinant enzyme and 10 mM G6P were used. Cells were subjected to serum starvation for 48 h. Error bars represent mean ± SEM. MW: molecular weight, FPLC: fast protein liquid chromatography, NT: no treatment, Veh: vehicle, WT: wild-type, Med: Mediterranean fibroblast

commonly used clinical therapy, but structural and functional changes in erythrocytes during storage, collectively referred to as storage lesion, remain concerned in transfusion practice[41]. Storage under conventional conditions is considered as oxidative stress for erythrocytes, as evidenced by increase in ROS over time and accumulation of oxidative biomarkers[42,43]. Thus, we determined that whether AG1 can improve preservation during refrigerated storage by monitoring the degree of hemolysis over 28 days. We found that AG1 (1 μM) reduces hemolysis over time by an average of 12% at day 28 (Fig. 5f and Supplementary Fig. 7b). Accordingly, the protein leakage from the treated erythrocytes was decreased as well (Fig. 5g and Supplementary Fig. 7c), which corresponded with increased G6PD activity (Fig. 5h). These data suggest that AG1-like compounds can serve as a novel preservative for prolonged storage of erythrocytes and impact to a broader population as well as G6PD-deficient patients.

## Discussion

In humans, in addition to the role of G6PD in preventing hemolysis (erythrocyte lysis), the anti-oxidant property of G6PD may relate to development of a variety of other pathologies, including kidney injury, heart failure, psychiatric disorder, diabetes, cholelithiasis, and cataract[44–52], suggesting that G6PD deficiency can be an underestimated risk factor for multiple human pathologies. Because AG1 increased the impaired activity of several common G6PD variants, our study suggests that a single pharmacological agent may provide treatment for several major G6PD enzymopathies, affecting many millions of people worldwide. Such a drug may also help prevent or reduce the sequelae of G6PD deficiency and/or synergize with other palliative treatments such as illumination for kernicterus[53,54]. We expect many other pathologies associated with G6PD deficiency, as aforementioned, to be affected by such treatment as well. Treatment with AGs may also be beneficial to G6PD-deficienct populations in developing countries, in which the use of hemolytic crisis-triggering drugs, like anti-malarial drugs (primaquine and chloroquine), are still common. We believe that AG1-like drugs may also be useful for preservation of blood for transfusion, as evidenced in Fig. 5 and for subjects with WT G6PD with other diseases associated with increased oxidative stress.

Human studies demonstrate that clinical pathology related to G6PD deficiency, at least as reflected by hemolytic crisis, occurs in subjects who carry a variant with <60% activity relative to the subjects with WT G6PD[55]. Therefore, although AG1 is safe, an optimal AG should be improved, to increase the catalytic activity in G6PD-deficient subjects to at least 60% of normal. Nonetheless, based on our initial studies, AG1 should be viewed as a

lead compound to correct G6PD deficiency and also to treat subjects with WT G6PD at risk of oxidative stress. Our current effort focuses on improving the biochemical and pharmacological features of AG1 through further medicinal chemistry efforts and structural studies. Finally, identifying small-molecule enzyme activators is still considered a challenging task in the field of drug discovery and development[56], and even less common is the ability to identify a small molecule that corrects functional defect due to different mutations; AG1 represents a compound that does both.

## Methods

**Materials.** Antibodies used in this study were principally purchased from Santa Cruz Biotechnology (G6PD (G-12): SC-373886 (dilution 1:1000), 6PGD (G-2): SC-398977 (dilution 1:1000), His (HIS.H8): SC-57598 (dilution 1:1000), Enolase (H-300): SC-15343 (dilution 1:1000)), Cell Signaling Technology (beta actin (8H10D10): 3700 S (dilution 1:1000)), Everest Biotech (G6PD: EB07841 (dilution 1:1000)), Advance Immunochemical (GAPDH (6C5): 6C5 (dilution 1:2000)), and Abcam (G6PD: AB87230 (dilution 1:1000), band 3 (EPR1426): AB108414 (dilution 1:1000)). TALON Superflow (28-9575-02) and bovine thrombin (91-030) were purchased from GE Healthcare and BioPharma for protein purification. *Pfu* Turbo polymerase (600252) used for site-directed mutagenesis was purchased from Agilent Technologies. Chymotrypsin was purchased from Promega (V1061). Cell counting kit-8 (CK04) and glutathione quantification kit (T419) were purchased from Dojindo. Cycloheximide (C7698) and chloromethyl-2′,7′-dichlorodihydro-fluorescein diacetate (CM-H$_2$DCFDA, C6827) were purchased from Sigma and Thermo Fisher Scientific, respectively. Glutathione assay kit (703002) used for blood assays was purchased from Cayman Chemical.

**Cell culture.** Lymphocytes derived from a normal subject (HG 00866) and a G6PD-deficient subject carrying the Canton variant (HG 02367) were purchased from Coriell Institute and cultured in RPMI 1640 supplemented with 15% fetal bovine serum (FBS), 100 U mL$^{-1}$ penicillin, and 100 μg mL$^{-1}$ streptomycin. An SH-SY5Y neuroblastoma cell line (ATCC CRL-2266) was cultured in Dulbecco's Modification of Eagle's Medium/Ham's F-12 50/50 Mix supplemented with 10% FBS, 100 U mL$^{-1}$ penicillin, and 100 μg mL$^{-1}$ streptomycin. A fibroblast cell line derived from a G6PD-deficient subject carrying the Mediterranean variant and normal fibroblast cell line as control were purchased from Coriell Institute (GM 01152) and Thermo Fisher Scientific (C0135C), respectively, and cultured in minimum essential medium supplemented with 15% FBS, 100 U mL$^{-1}$ penicillin, and 100 μg mL$^{-1}$ streptomycin. All the cell lines were maintained at 37 °C in a humidified incubator with an atmosphere of 5% of CO$_2$ and 95% air.

**Plasmid construction and site-directed mutagenesis.** The gene encoding WT G6PD was inserted into the pET-28a vector, using NdeI and SalI restriction enzyme sites. Site-directed mutagenesis was performed to generate G6PD variants using appropriate primer sets (Supplementary Table 1) according to the manufacturer's guidelines (Agilent-Quikchange site-directed mutagenesis). Briefly, the PCR reaction of 50 μL contained 10–50 ng of template, 125 ng of primers, 200 μM dNTPs, and 2.5 units of *Pfu* Turbo DNA polymerase. The reaction was initiated at 95 °C for 30 s to denature the template DNA, followed by 18 amplification cycles (95 °C for 30 s, 55 °C for 1 min and 68 °C for 7 min). All constructs were verified by sequencing.

**Protein expression and purification.** G6PD and its variants were expressed in *E. coli* C43 (DE3). When the culture density reached an OD$_{600}$ of 0.5–0.6, 0.5 mM

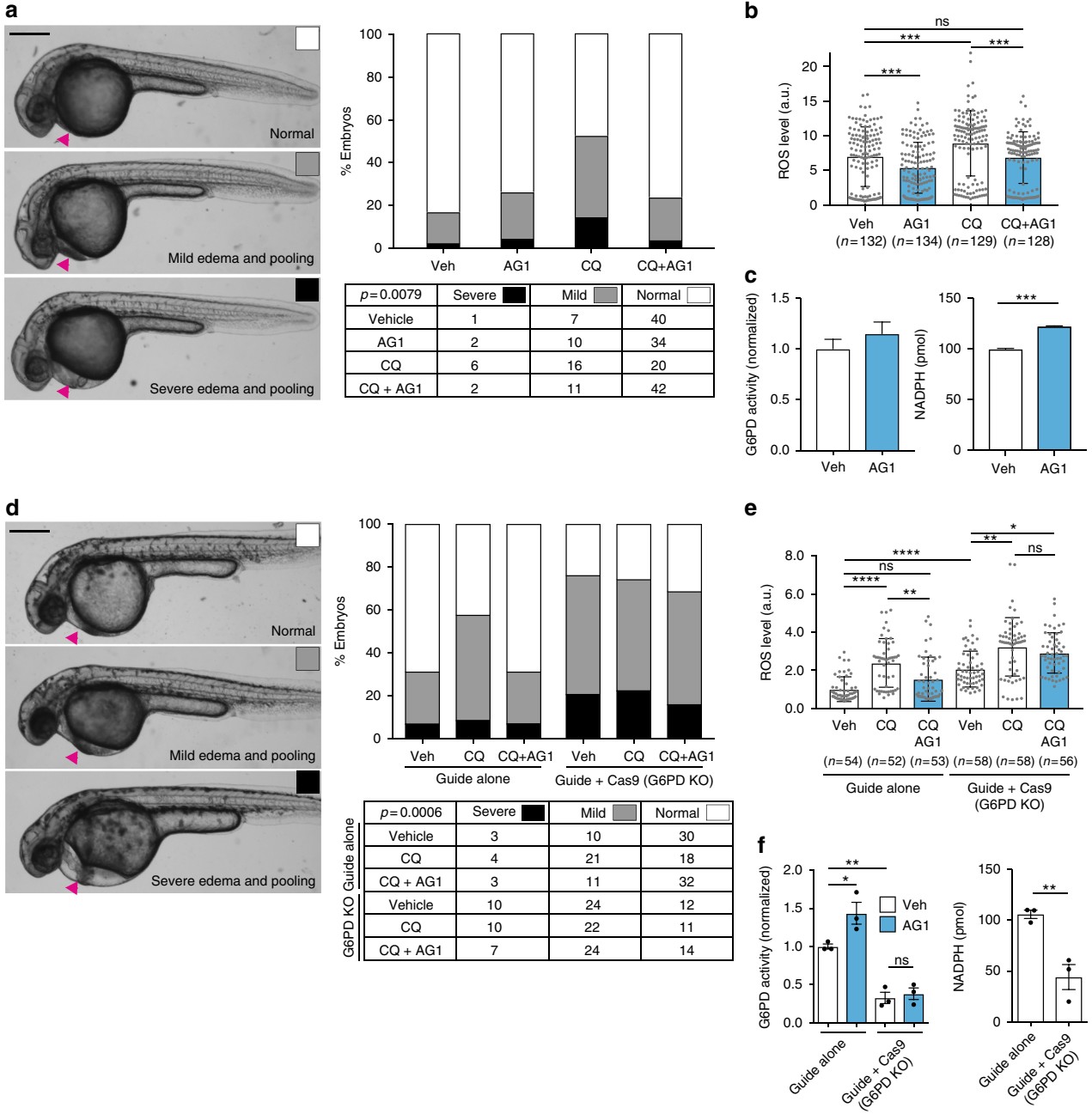

**Fig. 4** AG1 attenuates ROS-induced pericardial edema in a G6PD-dependent manner. **a** Embryos were treated at 24 hpf with 1 µM AG1 with and without chloroquine (CQ; 100 µg mL$^{-1}$) and scored at 32 hpf. Representative phenotypic images of pericardial edema and pooling (magenta arrows) are provided on the left (scale bar: 300 µm). Embryo orientation is lateral view, anterior left. Raw counts used for chi-square analysis and calculated p value are included in table below. **b** ROS levels in individual WT embryos from three independent clutches. Embryos were treated at 24 hpf for 5 h before ROS measurement. Error bars represent mean ± SD ($^{***}p < 0.001$, ns = not statistically significant, $p > 0.99$, Kruskal–Wallis multiple comparison test, adjusted $p$ value using Dunn's test). **c** G6PD activity and NADPH levels were measured using the lysates of pooled embryos (from two independent clutches). Error bars represent mean ± SEM ($^{***}p = 0.0003$, two-tailed unpaired Student's $t$-test). **d** Embryos were injected with either sgRNA targeting exon 10 of *g6pd* (Guide alone) or sgRNA + Cas9 protein (Guide + Cas9, G6PD KO (knockout)) to generate G6PD F0 crispants. Representative phenotypic images of pericardial edema and pooling (magenta arrows) are provided on the left (scale bar: 300 µm). Treatment conditions are the same as in **a**. Raw counts used for chi-square analysis and calculated p value are included in table below. **e** ROS levels in individual embryos with sgRNA or sgRNA + Cas9 (G6PD KO) protein injection. Treatment conditions and the statistics are the same as in **b** ($^{*}p = 0.0267$, $^{**}p < 0.01$, $^{****}p < 0.0001$, ns = not statistically significant). **f** G6PD activity and NADPH levels were measured with lysates of pooled embryos from three independent experiments. Error bars represent mean ± SEM of the replicate measurements ($^{*}p < 0.05$, $^{**}p < 0.01$, one-way ANOVA for G6PD activity measurement and two-tailed unpaired Student's $t$-test for NADPH measurement). Veh: vehicle, KO: knockout, CQ: chloroquine

IPTG was added to induce the protein expression. After culturing at 28 °C overnight, the bacteria were centrifuged, and the pellets were lysed by sonication in buffer containing 50 mM Tris (pH 7.4), 300 mM NaCl, 5% glycerol, 0.4 mM PMSF, 1 mg mL$^{-1}$ lysozyme, 0.1% Triton X-100 and protease inhibitor cocktail (Sigma P8340). G6PD was then purified by incubating the supernatant with TALON

Superflow resin equilibrated with 1 bed volume of equilibration buffer (50 mM Tris (pH 7.4), 300 mM NaCl and 5 mM imidazole) at 4 °C for 1 h. The resin was washed with 5 bed volumes of wash buffer (50 mM Tris (pH 7.4), 300 mM NaCl and 20 mM imidazole) in a gravity-flow column and resuspended in 5 mL of equilibration buffer for size exclusion chromatography (50 mM Tris (pH 7.4) and 150 mM

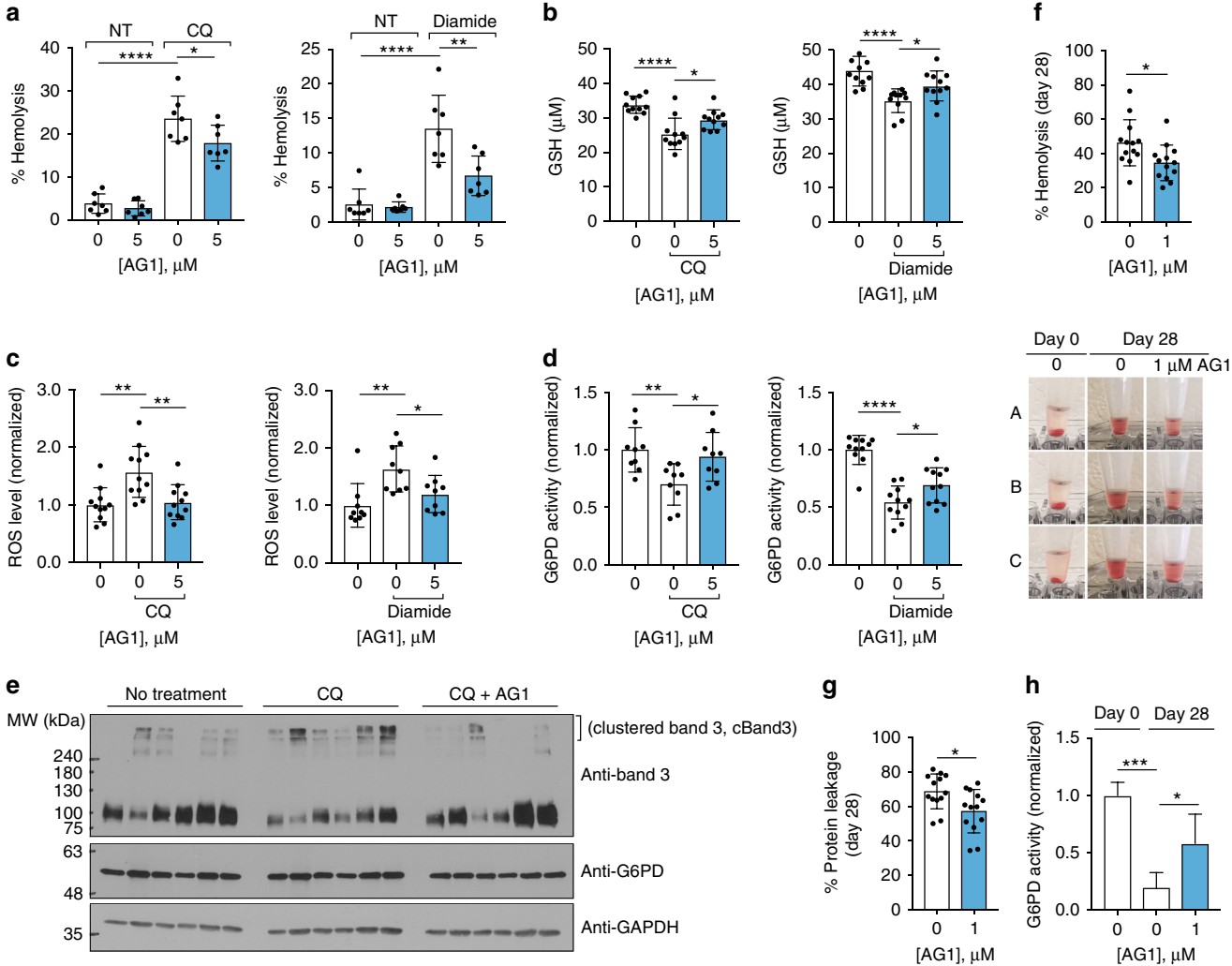

**Fig. 5** AG1 reduces hemolysis upon exposure to oxidative stressors. **a** AG1 reduced the extent of hemolysis of 5% erythrocyte suspension treated with either 1 mM chloroquine (CQ) for 4 h under light or 1 mM diamide for 4 h at 37 °C ($n = 7$ independent blood samples, $^*p = 0.0372$, $^{**}p = 0.0019$, $^{****}p < 0.0001$, one-way ANOVA). **b–d** AG1 significantly increased GSH levels and reduced ROS levels, when 5% erythrocyte suspension was exposed to either 1 mM chloroquine or diamide for 3 h at 37 °C, which was consistent with increased G6PD activity ($n = 9$–11, $^*p < 0.05$, $^{**}p < 0.01$, $^{****}p < 0.0001$, one-way ANOVA). **e** Band 3 protein was clustered (cBand3) when 5% erythrocyte suspension was treated with chloroquine, which was alleviated by AG1 treatment. Each lane represents one individual sample, and quantification is provided in Supplementary Fig. 7a. **f–h** AG1 (1 μM) improved storage of erythrocytes (5% suspension) at refrigerated temperature by reducing hemolysis and concomitant protein leakage for 28 days ($n = 13$ independent blood samples, $^*p < 0.05$, two-tailed unpaired Student's $t$-test), which corresponded with increased G6PD activity ($n = 4$, $^*p = 0.0323$, $^{***}p = 0.0003$, one-way ANOVA). Each sample was re-treated with AG1 every week. Representative hemolysis phenotypic images are provided below **f**. Error bars represent mean ± SD. NT: no treatment, CQ: chloroquine, cBand3: clustered band 3 protein

NaCl). Bovine thrombin (100 units) was added to the resin, which was followed by overnight incubation at 4 °C with gentle shaking. Tag-less G6PD was eluted and applied onto HiLoad 16/600 Superdex 75 pg size exclusion chromatography. Fractions containing G6PD were pooled and concentrated using 10 kDa MWCO membrane. The final concentration of G6PD was determined by Bradford method, and the protein was stored in 40% glycerol at −80 °C.

**Enzyme assay and kinetic measurements**. Enzyme activity was measured by monitoring NADPH production, which was coupled with diaphorase converting resazurin to fluorescent resorufin (excitation at 565 nm and emission at 590 nm) (Supplementary Fig. 3a); fluorescent signal was thus proportional to G6PD activity. All the assays were performed at 25 °C and run for 5 min in buffer containing 50 mM Tris (pH 7.4), 0.5 mM EDTA, 3.3 mM MgCl$_2$, 1 U mL$^{-1}$ diaphorase and 0.1 mM resazurin. 10 ng of recombinant enzymes or 10 μg of cell lysates was used for the assay with 10 μM NADP$^+$ (Sigma) as cofactor and 100 μM G6P (Sigma) as substrate. Steady-state kinetic parameters were obtained by varying concentrations of NADP$^+$ (0–10 μM) with a constant concentration of G6P (100 μM) and similarly for G6P (0–100 μM) with a constant concentration of NADP$^+$ (10 μM) with 10 ng of recombinant enzymes. Data analysis was performed using GraphPad

Prism software v.6 (GraphPad Software, La Jolla, CA USA). Kinetic parameters were obtained by fitting the data to the Michaelis–Menten equation.

**High-throughput screening assay**. The diaphorase-coupled enzyme assay as described above was used to screen for small-molecule activators. All the compounds purchased from SPECS, Chembridge, ChemDiv Kinase, LOPAC, Microsource Spectrum, Biomol FDA, Biomol ICCB, and NIH Clinical Collection were added to Canton G6PD enzyme reaction mixture at a final concentration of 16.67 μM using Caliper Life Sciences Staccato system with a Twister II robot and a Sciclone ALH3000 (Caliper Life Sciences, Alameda, CA USA) integrated with a V&P Scientific pin tool, which was followed by the incubation for 3 h. Then addition of G6P initiated the reaction, which was run for 2.5 min. The fluorescent signals were recorded 4 times during the run using Molecular Devices AnalystGT (Molecular Devices, Sunnyvale, CA USA). Any compounds showing 30% activation of the enzyme were rescreened in a dose-dependent manner (0–30 μM, duplicate) to identify potential hits. The screening and data analysis were carried out by Stanford University High-Throughput Bioscience Centers (HTBC).

**Thermostability assay**. 10 ng of recombinant enzymes was incubated at various temperatures ranging from 25 to 65 °C for 20 min, and the activity was measured as

described above. After normalization of the data between 0 and 100, Boltzmann sigmoidal equation was used to calculate the $T_{1/2}$ value, the temperature at which the enzyme retains half of original activity.

**In vitro proteolysis assay**. 200 ng of recombinant enzymes was incubated with 10 ng of chymotrypsin for 1 h at room temperature. 100 μM of the compound was added to some reaction conditions. The following protein level was examined by Western blot (original blot data are available in the Supplementary Information).

**Overexpression of WT G6PD and Canton variant in SH-SY5Y cells**. Prior to the cellular-based assays using SH-SH5Y cells, the duration of overexpression of WT G6PD and Canton variant was examined by transfecting the cells seeded in a 12-well plate. The genes encoding human WT G6PD and Canton variant were first PCR-amplified, which was then inserted into pcDNA 3.1/myc-His C (Thermo-Fisher Scientific) using HindIII and XhoI restriction enzyme sites. 0.5 μg of cDNA and 1.5 μg of lipofectamine (Invitrogen) were diluted in 50 μL of Opti-MEM medium, respectively and incubated for 5 min. The diluted DNA was combined with diluted lipofectamine, which was followed by incubation for 20 min prior to the addition to cells. The transfected cells were collected at different time points (up to 72 h), and the overexpressed G6PD levels were examined by Western blot. The transfection was carried out in 50% serum-starved cells. Once the duration of expression was confirmed, other cellular-based assays were performed.

**Cycloheximide chase assay**. 50,000 cells (SH-SH5Y cells or fibroblast cells) or 100,000 cells (lymphocytes) were seeded in a 12-well plate and incubated overnight. The cells were subjected to serum starvation (50–75%) for 48 h and treated with 50 μg mL$^{-1}$ of cycloheximide at different time points (0–48 h). The cells were treated with the compound (1 μM) for 48 h together with cycloheximide. Then the cells were collected in PBS containing protease inhibitor cocktail and 1% Triton X-100 and centrifuged at $18,800 \times g$ at 4 °C for 10 min. 10 μg of total protein was loaded onto SDS–PAGE gels, and the protein levels were examined by Western blot (original blot data are available in Supplementary Information).

**Glutathione (GSH) measurement**. Total glutathione level was measured using a Total Glutathione Quantification Kit (Dojindo), according to the manufacturer's instructions. Cells were subjected to serum starvation (50–75%) for 48 h to induce oxidative stress and treated with the compound for 24 h before measurement. The absorbance was read at 412 nm.

**Cell viability assay**. Cell viability was measured using a Cell-Counting Kit-8 (CCK-8, Dojindo), utilizing WST-8 [2-(2-methoxy-4-nitrophenyl)-3-(4-nitrophenyl)-5-(2,4-disulfophenyl)-2H-tetrazolium monosodium salt, according to the manufacturer's instructions. Cells were subjected to serum starvation (50–75%) for 48 h before measurement. Cells in 100 μL of medium per well were treated with 10 μL of CCK-8 solution and incubated for 2 h at 37 °C. The absorbance was read at 450 nm. Viability of lymphocytes was measured by staining with a 0.4% solution of trypan blue in buffered isotonic salt solution (pH 7.2). The viability was calculated as the number of viable cells (non-stained by the dye) divided by the total number of cells within the grids on the hemacytometer. Cells were treated with 1 μM of the compound 24 h before the measurement.

**Cellular reactive oxygen species (ROS) measurement**. Cells in a 96-well plate were incubated with chloromethyl-2′,7′-dichlorodihydrofluorescein diacetate (CM-H$_2$DCFDA) at a final concentration of 5 μM in HBSS (Hank's balanced salt solution) for 30 min at 37 °C. After wash, cells were treated with Hoechst 33342 (Molecular Probes) to stain nuclei and incubated for another 10 min at 37 °C. Cells were washed in HBSS, and the florescence was analyzed with excitation/emission at 485/525 nm. Cells were subjected to serum starvation (50–75%) for 48 h to induce oxidative stress and treated with the compound for 24 h before measurement. The signal was normalized to nuclei staining (excitation/emission at 350/470 nm).

**Native gel electrophoresis**. 200–300 ng of recombinant enzymes was incubated with the compound (varying in assays) for 10 min at room temperature. To prevent streaking and artifacts in native PAGE gel, the samples in a native state (no boiling of sample and no reducing agent in sample buffer) were electrophoresed by SDS–PAGE. Cross-linking assay was initiated by incubating 200 ng of recombinant enzymes in PBS with 0.1% of glutaraldehyde and different concentrations of NADP$^+$ (0, 10, 100, 1000 μM) or MgCl$_2$ (0, 1, 10, 100 mM). The mixture was incubated for 10 min at room temperature, which was followed by the addition of 100 mM Tris (pH 8.0) at a final concentration to terminate the reaction. The samples were electrophoresed as described above.

**G6PD siRNA-knockdown assay**. Endogenous G6PD was knocked down in each cell line as follows; 2 pmol of siRNA G6PD (Santa Cruz Biotechnology, sc-60667) and 0.5 μg lipofectamine (Invitrogen) were diluted in 10 μL of Opti-MEM medium, respectively and incubated for 5 min. The diluted siRNA was combined with

diluted lipofectamine, which was followed by additional incubation for 20 min prior to the addition to cells in a 96-well plate.

**Crystallization, data collection, structure determination and refinement**. Crystals of WT G6PD recombinant enzyme grew in sitting drops containing 20% w/v PEG 3350, 0.2 M potassium formate, pH 7.3. Suramin (G6PD inhibitor) was added to the protein solution (the final concentration in the drop was 0.5 mM) prior to the crystallization. Canton G6PD recombinant enzyme was crystallized in sitting drops containing 20% w/v PEG 3350, 0.2 M ammonium citrate tribasic, pH 7.0. AG1 dissolved in 30% DMSO was added to the protein prior to the crystallization to reach final concentration of 0.5 mM in the drop. None of the Suramin or AG1 compound was visible in the electron density map of WT G6PD and Canton G6PD; however, they significantly improved diffracting quality of the crystals. Note that our inability to observe AG1 in the crystal structure may reflect instability or flexibility of the ligand bound to G6PD. Additional crystallographic studies and further medicinal chemistry efforts will help determine the binding site of AG1 in the enzyme and the mechanism by which it activates G6PD. X-ray diffraction data of WT G6PD and Canton G6PD were collected at 100 K at beamline 12–2 of Stanford Synchrotron Radiation Light Source (SSRL) and beamline 5.0.2 of Advanced Light Source (ALS), respectively. A solution of 20% glycerol was used as cryo-protectant. Crystals of WT and Canton G6PD diffracted to 1.9 Å and 2.6 Å resolution, respectively. The data were processed using iMOSFLM[57], and further analysis of the data by POINTLESS[58,59] indicated the space group of F222 and $P2_12_12_1$ for WT G6PD and Canton G6PD crystals, respectively. WT G6PD structure was solved using molecular replacement with a monomeric G6PD structure from PDB: 2BHL used as a search model in MOLREP. Canton G6PD structure was solved using the WT G6PD structure that was already solved in this study. Molecular models were further built in Coot[60]. Both structures were refined using the restrained isotropic refinement in REFMAC[61,62]. TLS parameters were not used for the refinement in both cases. Each refinement was done using 10 cycles of maximum likelihood restrained refinement, with geometry weight adjusted to 0.05. Data collection and refinement statistics are summarized in Table 1. The atomic coordinates and structure factors are deposited in the PDB database under accession codes PDB: 6E08 for WT G6PD and PDB: 6E07 for Canton G6PD. All structure figures were prepared using PyMOL (PyMOL Molecular Graphics System, Version 1.5.0.5; Schrödinger).

**Zebrafish husbandry**. Adult zebrafish (AB strain; 3–18 months old) were raised according to standard protocols, and embryos were obtained through natural mating and staged[63]. Adult $tbx16^{b104/+}$ was a generous gift from S. Amacher (Ohio State University). All animal procedures were performed according to NIH guidelines and approved by the Committee on Administrative Panel on Laboratory Animal Care (APLAC) at Stanford University. Embryos were raised in E3 medium at 28.5 °C.

**Zebrafish crispants**. sgRNA against exon 10 of g6pd was designed using CHOPCHOP and in vitro transcribed using the T7 quick high yield RNA synthesis kit (New England Biolabs, E2050S)[64,65]. The gene-specific oligo sequence was: 5′-T AATACGACTCACTATAGAGAAGGGGAGGCAAAACTGGTTTTAGAGCTAG AAATAGCAAG-3'. sgRNA and Cas9 protein (New England Biolabs, M0386T) were mixed and microinjected into one-cell-stage embryos. For each injected clutch, 10 individual embryos were isolated at 24 hpf for sequencing to confirm introduction of a CRISPR-mediated indel in exon 10.

**Zebrafish compound treatment**. Embryos were dechorinated with pronase at 24 hpf and treated with 100 μg mL$^{-1}$ of chloroquine and/or AG1 (1 μM) by directly adding the compounds to the well. For ROS measurements, the embryos were incubated with compounds for 5 h and then the ROS-detecting reagent (CM-H$_2$DCFDA) was added at a final concentration of 500 ng mL$^{-1}$ to the well and incubated for 3 h. One embryo was placed to each well of a black, opaque 96-well plate. The florescence was analyzed with excitation/emission at 485/525 nm. After ROS assays, embryos at about 32 hpf were pooled and lysed in buffer containing 50 mM Tris (pH 7.4), 150 mM NaCl, 1 mM EDTA, 0.1% NP-40 and protease inhibitor cocktail, which was followed by three cycles of freeze and thawing in liquid nitrogen. The lysate was centrifuged at $18,800 \times g$ at 4 °C for 15 min. Total protein concentration in the supernatants (total lysate) was determined by the Bradford method. 10 μg of total lysate was used for enzymatic assay. 50 μg of total lysate was used to measure NADPH levels using a NADPH quantification kit (Biovision), according to the manufacturer's instructions. 50 μg of total lysate was loaded onto 10% SDS–PAGE gels, and the protein levels were examined by Western blot using anti-G6PD antibody (Abcam (G6PD: AB87230)).

**Zebrafish imaging**. Live embryos were anesthetized and mounted in 3% methylcellulose. Embryos were imaged with a Leica M205FA microscope equipped with a 1.0x Plan Apochromatic objective and a SPOT Flex camera or a Leica DM4500B compound microscope equipped with a QImaging Retiga-SRV camera. For hemoglobin staining, live embryos were stained in 0.6 mg mL$^{-1}$ o-dianisidine solution containing 10 mM sodium acetate (pH 4.5), 0.65% H$_2$O$_2$ and 40% ethanol in the dark for 15 min and cleared in glycerol[37]. Embryos were then mounted in 100% glycerol

and imaged with a Leica M205FA microscope equipped with a 1.0x Plan Apochromatic objective and a SPOT Flex camera. All images were captured using SPOT or MetaMorph imaging software (Diagnostic Imaging Inc.) and processed in Photoshop (Adobe). Adjustments were limited to brightness levels and cropping. Analysis was carried out by an observer blinded to the experimental conditions.

**Blood sample assay**. De-identified blood samples were obtained from the Stanford Blood Center. Erythrocytes were collected by filtering the samples through a cellulose slurry to remove platelets and leukocytes and then washed with saline. G6PD activity was measured spectrophotometrically by the Beutler method[66]. The activity of all the samples used in this study was in a normal range (5–9 U g$^{-1}$ Hb), suggesting that the subjects have WT G6PD. 5% erythrocyte suspension was pre-incubated with 1–5 μM AG1 at 4 °C overnight, which was followed by treatment with (or without) either 1 mM chloroquine (CQ) or 1 mM diamide for 3–4 h at 37 °C (for hemolysis assay with chloroquine, the mixture was incubated under light). Then centrifugation at $100 \times g$ for 5 min was followed. Hemoglobin release in the supernatant was monitored by measuring absorbance at 540 nm. Saline was used as a negative control (0% hemolysis) and a sample treated with 0.1% Triton X-100 was used as a positive control (100% hemolysis). For ROS measurement, erythrocyte mixture was washed with saline by centrifugation after treatment and incubated with chloromethyl-2′,7′-dichlorodihydrofluorescein diacetate (CM-H$_2$DCFDA) at a final concentration of 5 μM in saline at 37 °C for 15 min. After wash, the samples were lysed with 0.1% Triton X-100 (final concentration), and the florescence was analyzed with excitation/emission at 485/525 nm. GSH measurement was determined using a Cayman glutathione assay kit (Cayman Chemicals, 703002). Briefly, 50 μL of diluted erythrocyte lysate samples were mixed with 150 μL of assay reagents including glutathione reductase, 5′,5-dithiobis-(2-nitrobenzoic acid) (DTNB) and NADPH, which was followed by incubation for 25 min at room temperature. The absorbance was read at 412 nm. For storage assay, 5% erythrocyte suspension was stored at 4 °C with and without 1 μM AG1, and hemolysis and G6PD activity were monitored every week for 28 days to examine whether AG1 improves preservation of erythrocytes over time. Protein leakage was also examined by measuring the absorbance of the supernatant of samples at 280 nm. The samples were re-treated with AG1 every week.

**Statistical analyses**. Most assays were repeated at least in three independent experiments. The data from in vitro and cell-based assays are presented as mean ± standard error of the mean (SEM), and the data from the human erythrocyte study are presented as mean ± standard deviation (SD). Statistical differences were calculated by either Student's *t*-test (two-tailed, unpaired), one-way ANOVA, or two-way ANOVA using GraphPad Prism software. In assays with human erythrocytes, each sample was utilized as its own control, and assay parameters were compared before and after treatment; thus, randomization was not needed. For all zebrafish experiments, at least two breeding tanks, each containing 3–4 males and 3–5 females from separate stocks, were set up to generate embryos. Embryos from each tank were randomly distributed across tested conditions. Unfertilized and developmentally abnormal embryos were removed prior to compound treatment. No statistical methods were used to determine sample size per condition. For phenotypic analysis, raw counts for each condition were used for chi-square analysis or Fisher's exact test based on expected values. For all phenotypic analysis, the scorer was blinded to treatment conditions. For ROS assays, the normality of each distribution was assessed using the Shapiro–Wilk test and determined to be non-normal. A Kruskal–Wallis test with a Dunn's secondary test was used to determine differences between all conditions. *p* values were corrected for multiple comparisons testing. *p* values and number of samples or experiments replicated are indicated within the figure legends, and $p < 0.05$ was considered statistically significant.

## Data availability

The data that support the findings of this study are available from the corresponding author upon request. Protein structures have been deposited in the Protein Data Bank (PDB) with accession codes of 6E07 and 6E08 for Canton G6PD and WT G6PD, respectively.

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

## Acknowledgements

This paper is in memory of Professor Adrienne Gordon, the original AG, who advised us on this project and many past ones. We thank David E. Solow-Cordero for the high-throughput screen, Anna Cunningham and Bereketeab Haileselassie for helpful discussions, former and current D.M.-R. lab members for their generous support, Carolyn Wong and Professor Bertil Glader for their advice on RBC studies. The study was supported by a National Institutes of Health (NIH) grant HD084422 to D.M.-R. and seed funds from the NIH National Center for Advancing Translational Science Clinical and Translational Science Award (UL1TR001085), SPARK (a translational research programme at Stanford), Weston Havens Foundation, and the Stanford Child Health Research Institute Transdisciplinary Initiatives (TIP) grant. S. Hwang was supported by the Stanford Child Health Research Institute postdoctoral grant support award, and K. Mruk was supported by a Craig H. Neilsen postdoctoral fellowship.

## Author contributions

C.-H.C. and D.M.-R. conceived the project. S.H. and D.M.-R. designed the study and wrote the manuscript with inputs from all authors. K.M. performed experiments in zebrafish under the guidance of J.K.C., S.R. and N.H. performed structural studies under the guidance of S.W. A.G.R. contributed to structural studies and L.E.D. contributed to cell-based assays. All authors discussed the results and commented on the manuscript.

## Additional information

**Competing interests:** The authors declare no competing interests.

