## [Peer Review File · Nature Communications]

Reviewers' Comments:

Reviewer #1:

Remarks to the Author:

The manuscript by Hwang et al describes the discovery of a small molecule activator of G6PD variants implicated in the pathology of G6PD deficiency disorder. Their findings constitute an important step towards a pharmacological approach for the treatment of G6PD deficiency and associated enzymopathies and should be published. The structural work is mostly technically sound and the conclusions drawn are adequately supported by the data presented. However, it is surprising that AG1 has not been found bound at the crystal structure and the authors need to discuss this, especially since glycerol and NADP (not included in the crystallization media) were found bound at the protein. Furthermore, the authors need to mention the refinement procedure used within REFMAC (has TLS been used and how?) for both structures since the ratio of refined parameters to the number of reflections appears to be disturbingly low.

Reviewer #2:

Remarks to the Author:

Correcting glucose-6-phosphate dehydrogenase (G6PD) deficiency with a small molecule activator, by Sunhee Hwang et al, Daria Mochly-Rosen, CA

G6PD deficiency, the most common human enzyme deficiency, is caused by over 160 different point mutations. As G6PD is a major source of protective anti-oxidants through NADPH production, G6PD deficiency may result most frequently in hemolytic anemia and other damage associated with increased oxidative stress. Using high-throughput screening, the Authors identified AG1, a small molecule that increases the activity of the wild-type, the Canton G6PD-deficient mutant and several other common human G6PD mutants. AG1 reduced oxidative stress and increased NADPH levels in the zebrafish G6PD-deficiency model, and prevented chloroquine-induced hemolysis of human erythrocytes (RBC). G6PD is essential in preserving the integrity of RBC because they have no other NADPH-generating enzymes to protect against oxidative stress. The Authors suggest that a pharmacological agent, of which AG1 may be a lead, may alleviate the challenges associated with G6PD deficiency.

This is a clearly written, interesting and novel paper of potential interest for the vast community of the >400 mio G6PD-deficient subjects worldwide. As detailed below, however, there are some major problems.

1. The Authors describe a recent study (Long Chen et al: Data mining and pathway analysis of glucose-6-phosphate dehydrogenase with natural language processing, *Mol Med Rep.* 2017;16:1900-1910, ref. 4) that points at the potential involvement of G6PD and G6PD-deficiency in various cancer types, autoimmune diseases, and oxidative stress induced disorders. Based of those studies, the Authors feel that AG1 (their G6PD-activity-increasing-molecule) may be the lead of a novel class of drugs able to increase G6PD-activity and so avert the widespread health risks connected with the deficiency. A number of studies performed in Sardinia where 12-15% of males is almost completely G6PD-deficient (G6PD-Med variant with <3-5% residual activity) indicate that the deficiency is perfectly well compensated (except for favism and kernicterus in infancy).

Therefore, present reviewer considers the prospect of a pharmacological "therapy" and "correction" of G6PD-deficiency as unrealistic for the following reasons:

a) Life expectancy of G6PD-deficient subjects is significantly higher compared to normal subjects. For example, a cohort of G6PD-deficient men (Mediterranean variant) was followed during 11 years and compared with a normal G6PD cohort. At the trial end, the deficient cohort experienced a significantly reduced overall mortality mostly due to four-fold reduction of cerebrovascular and heart disease.

b) The prevalence of Sardinian G6PD-Med-deficient male centenarians was four-fold that of continental Italians (a, b: see AG Schwartz, LL Pashko, *Aging Res Rev.* 2004;3:171-187).

- c) Clinical and mortality studies indicate that cardiovascular disease susceptibility (CDS) was significantly lower in G6PD-Med-deficient men (see Manganelli et al, *Cardiovasc Hematol Dis.* 2013; 13: 73-82).
- d) The risk of colorectal cancer was significantly reduced in G6PD-Med-deficient men (Dore et al, *Medicine.* 2016; 95: 44) and suppression of G6PD-activity by DHEA (a powerful G6PD-inhibitor) showed inhibition of cervical cancer (Fang Z et al, *BBActa* 2016; 1863: 2245-2254). In more general terms though, contradictory results on the relationships between G6PD-deficiency and cancer are present in the literature (Cocco P, *J Epidemiol Community Health.* 1987; 41(2):89-93).
- e) By contrast with a)-d), published evidence supports the positive correlation between G6PD-deficiency and diabetes (see Meloni et al, *Haematologica*, 1992; 77:94-95; Heymann et al, *Diabetes Care* 2012; 35:e58), cataract (see Moro et al, *Am J Hum Genet* 1985; 5: 197-200) and cholelithiasis (see Meloni et al, *Acta Haematol* 1991; 85: 76-78).
- f) Contrary to the present Authors' suggestion to treat deficient subjects with G6PD-activity enhancers, the exact opposite has been suggested by others, namely to block G6PD-activity aimed at alleviating G6PD-deficiency-dependent pathologies. Rationale and prospects of this suggestion, based on administration of dehydroepiandrosterone (DHEA), a powerful inhibitor of G6PD, is summarized in a review article by Schwartz and Pashko (*Ageing Res Rev* 2004; 3: 171-187).
- g) Lastly, G6PD-deficiency provides distinct resistance against severe malaria. Making G6PD-deficient subjects pharmacologically normal would obliterate such essential mechanism of anti-malaria defense. Interestingly, the triad: falciparum malaria, G6PD-deficiency and fava bean consumption was present in almost all (except sub-Saharan Africa where fava beans were not cultivated) malarial regions. As discussed below, vicine and convicine of faba beans elicit hemolysis ("favism") by the same mechanism as chloroquine.

2. The Authors correctly emphasize the inability of the G6PD-deficient RBCs to keep glutathione in the reduced state upon oxidative stress, and more generally the propensity of deficient RBC to oxidative hemolysis (and kernicterus). In fact those are the most frequent severe pathologies observed in G6PD-deficiency.

Specifically, G6PD-deficient subjects are at risk of severe hemolysis upon administration of primaquine, an essential drug for treatment of vivax malaria. Origin and implications of G6PD-deficiency and primaquine toxicity in vivax malaria are well described in K. Baird's review article in *Pathogens and Global Health*, 2015; 109: 93-106.

The present Authors underscore the importance of their suggested "therapy" of G6PD-deficiency, emphasizing the reversal (or abrogation) of primaquine-elicited hemolysis through increase in the activity of RBC G6PD.

Evidence and discussion of reversal of PQ hemolysis though are not satisfactory for the following reasons:

- a) Primaquine (PQ) hemolysis and favism, ie hemolysis after consumption of fava beans (FB) in G6PD-deficient subjects, are superimposable. Both are characterized by the fast, profound and irreversible oxidation of GSH in the RBC. GSH oxidation starts a chain of molecular events that finally lead to massive phagocytic elimination of RBC and anemia.
- b) While the molecular details of PQ hemolysis are scarce, favism hemolysis has been described in detail (see for example Luzzatto L, Arese P. Favism and glucose-6-phosphate dehydrogenase deficiency. *N Engl J Med.* 2018 Jan 4; 378(1):60-71; Arese P, Pathophysiology of hemolysis in glucose-6-phosphate dehydrogenase deficiency. *Semin Hematol*, 1990; 27: 1-40).
- c) In summary, primaquine or vicine and convicine, the two redox-active compounds present in very high concentrations in FB, oxidize GSH in G6PD-deficient RBC. Due to insufficient NADPH production, the NADPH-dependent enzyme glutathione reductase (the GSH-regenerating enzyme) does not work. For this reason, reduced glutathione is not regenerated. Low GSH does not allow RBC protection against ROS radicals and hydrogen peroxide continuously generated in the RBC; ROS radicals and peroxide oxidize RBC protein thiols and lipids, generate the powerful oxidants ferryl Hb and hemichromes and cause iron release from Hb. RBC membrane proteins are oxidized and clustered, generating Heinz bodies. Finally, clustered band 3 leads to membrane deposition of autologous IgG and complement, leading to RBC phagocytosis and extravascular hemolysis (EH). EH is the vastly predominant form of RBC destruction in favism and primaquine hemolysis.

d) Importantly, if deficient G6PD was indeed activated by AG1 treatment (as asserted by the Authors), we should expect a remarkable GSH regeneration in the isolated, deficient RBC pre-treated with a specific GSH oxidant, such as FB components or diamide or similar GSH oxidants. This essential proof of evidence for AG1-induced G6PD activation and GSH regeneration is sorely missing.

e) In conclusion, there is no evidence that AG1 works and re-activates the mutant G6PD.

Reviewer #3:

Remarks to the Author:

Hwang and colleagues and colleagues have explored the activity of Canton G6PD, a specific mutant found in Asia. The authors first performed biochemical characterization of Canton G6PD evaluating protein stability and enzyme activity. Next, they determined the structural changes in Canton G6PD (Figure 2). Using a biochemical screen, the authors next determined that a novel compound AG1 could increase Canton G6PD activity and induce biochemical changes. Using a zebrafish model, the authors showed the rescue of hemolysis and ROS from chloroquine exposure using AG1. G6PD knock down using Cas9 did not show activation as proof of specificity. Finally, using the red cell "storage lesion" as a model, the authors found improvements in protein leakage and G6PD activity.

Overall this is a well-written concise manuscript with novel findings and possible translation into the clinical realm. There is very little work in the field of small molecule discovery for G6PD deficiency, making this work novel, important to publish, and likely high impact.

A have no concerns about the manuscript or experiments shown.

Comments:

Did the authors have access to Canton G6PD (or A-) blood to test AG1?

In the phenotyping of AG1 treated zebrafish, can authors show statistical improvement in the phenotype (Chi-Square)?

How did the authors arrive at 1 μ M concentration to be used in the zebrafish testing?

Was there a death phenotype in the zebrafish experiments and did AG1 rescue it?

Reviewer #1:

The structural work is mostly technically sound, and the conclusions drawn are adequately supported by the data presented. However, it is surprising that AG1 has not been found bound at the crystal structure and the authors need to discuss this, especially since glycerol and NADP (not included in the crystallization media) were found bound at the protein.

R: Thank you for your positive feedback. Like the reviewer, we were puzzled by why AG1 was not observed in the original analysis. In addition to noting on that in the manuscript (p.20), we provide a more detailed discussion of our current mechanistic understanding of AG1 below. This represents an ongoing effort in our labs, and the data from these studies support a mechanism whereby AG1 activates G6PD by promoting oligomerization of G6PD to the catalytically competent form(s). As this effort is incomplete, we feel that it is premature to include it in this first report on AG1-induced biochemical and biological characterization.

Attempts of visualizing AG1: soaking G6PD crystals with AG1

We used another crystallization method for G6PD in complex with AG1. Based on our current mechanistic understanding, we corrected the structure originally provided by our HTS facility to the active species (AG1), which is now commented on p. 8 in the manuscript. The active species is a molecule connected *via* a disulfide bridge, which was confirmed by mass spectroscopy (see Supplementary Figure 3a). This oxidation occurs under ambient conditions and similar observations have been made elsewhere in the literature for primary thiols (see Hebeisen, P. *Bioorganic Med. Chem. Lett.* 2008, 18, 4708–4712 for a relevant example).

When this active species was independently synthesized and purified, it had the same activity as the sample provided by our HTS facility across a variety of assays. After preparing new batches of crystals of the wild-type G6PD using the condition containing 0.1 M sodium citrate (pH 5.4), 50~100 mM glycolic acid, 6~10% PEG3350, and 2 mM NADP⁺, crystals were soaked in the crystallization buffer containing 5 mM AG1 for two days. We collected a new diffraction dataset at 2.93 Å resolution from one of these crystals at the BL12-2 beamline of the Stanford Synchrotron Radiation Lightsource (SSRL). The dataset was processed using the HKL2000 and CCP4 program. The space group of the new crystals was found to be C222₁ with one tetramer in the asymmetric unit (ASU), as opposed to the F222 (WT) with one tetramer in ASU and P2₁2₁2₁ (Canton) with two tetramers in ASU for our structures described in our manuscript.

The structure of G6PD in the presence of AG1 was determined by molecular replacement with the MOLREP program, using the crystal structure of the G6PD Canton mutant monomer as the search model. The refinements and model building of the atomic coordinates were performed using the PHENIX and Coot programs. The electron density around the dimer interface was shown at different sigma levels in Figure 1 below. The electron density for both indole rings and the disulfide bond of AG1 is observed, but that for carbon atoms between the indole ring and the sulfur atom is poor and only visible at lower σ -levels (Figure 1). We tried occupancy refinement of AG1, which resulted in 0.8 to 0.9, depending on the initial occupancy, but the density maps appear similar, i.e. rugged AG1 electron density. Interestingly, the electron density map showed that one Trp509 is flipped out (Figure 1). The reason for the weak electron density of AG1 may be due either to the instability or flexibility of AG1 molecules bound to G6PD. Hence, we didn't deposit this structure to the Protein Data Bank.

[2mFo-DFc map]

Figure 1 – The close-up view around the interface of the G6PD dimer between the structural NADP⁺-binding sites. G6PD molecules are shown in magenta and cyan, the structural NADP⁺ molecules in green, and AG1 molecule in yellow. The 2mFo-DFc map was calculated with AG1 occupancy of 0.84 and 0.79 (the other side, not shown), and contoured at the 1.0, 0.8, 0.6, and 0.4 σ levels.

Furthermore, mutational analyses around the two C₂ symmetric regions of Canton G6PD support the observations made in the crystal structure presented above. Residues complementary to the ligand on the C₂ symmetric site between the structural NADP⁺-binding sites (binding site supported by crystallography) were mutated to determine their effect on the AC₅₀ of AG1. The ratio of the AC₅₀ of the ligand in the relevant mutation to the AC₅₀ of the ligand in Canton G6PD is provided in Figure 2, below. Value greater than 1 indicates that the mutation attenuated the activity of AG1 (weakened binding). Two anionic residues close to the symmetry axis, Asp421

and Glu419 were mutated to alanine individually and as a double mutation. Asp421A and the double mutation, Asp421A/Glu419A, significantly weakened the AC_{50} of AG1, consistent with the crystallography study. Conversely, mutations made distal to the dimer interface between the structural $NADP^+$ -binding sites (G222A, D228A and D347A) had no effect on AG1 activation of G6PD. Taken together, these data support binding of AG1 to the dimer interface between the structural $NADP^+$ -binding sites.

Figure 2 – Mutational analysis. (Left) Surface of G6PD indicating the two locations about the C_2 symmetry axis where mutations were made. (Right) Mutations near the structural $NADP^+$ -binding site significantly affect the activity of the activator AG1.

As this crystallographic study together with further medicinal chemistry efforts (SAR) is ongoing and outside the immediate focus of this study, we hope that our choice not to include the above data in the current manuscript is acceptable. Instead, we added a sentence of “Note that our inability to observe AG1 in the crystal structure may reflect instability or flexibility of the ligand bound to G6PD. Additional crystallographic studies and further medicinal chemistry efforts will help determine the binding site of AG1 in the enzyme and the mechanism by which it activates G6PD” to the Method section (p.20).

Furthermore, the authors need to mention the refinement procedure used within REFMAC (has TLS been used and how?) for both structures since the ratio of refined parameters to the number of reflections appears to be disturbingly low.

R: The WT and Canton G6PD structures were refined using the restrained isotropic refinement in REFMAC. TLS parameters were not used for the refinement in both cases. Each refinement was done using 10 cycles of maximum likelihood restrained refinement, with geometry weight adjusted to 0.05 (now added to the Method section, p.20).

Reviewer #2:

This is a clearly written, interesting and novel paper of potential interest for the vast community of the >400 million G6PD-deficient subjects worldwide.

As detailed below, however, there are some major problems.

R: We thank the reviewer for the thoughtful comments regarding the risk and benefit profile of G6PD deficiency in a variety of disease processes. While the detrimental effects of G6PD deficiency are clear in favism and kernicterus of infancy, more recent associations identified in various diseases have not been completely delineated. Please see our responses to some of the examples brought up by the reviewer below.

1. The Authors describe a recent study (Long Chen et al: Data mining and pathway analysis of glucose-6-phosphate dehydrogenase with natural language processing, *Mol Med Rep.* 2017;16:1900-1910, ref. 4) that points at the potential involvement of G6PD and G6PD-deficiency in various cancer types, autoimmune diseases, and oxidative stress induced disorders. Based on those studies, the Authors feel that AG1 (their G6PD-activity-increasing-molecule) may be the lead of a novel class of drugs able to increase G6PD-activity and so avert the widespread health risks connected with the deficiency. A number of studies performed in Sardinia where 12-15% of males is almost completely G6PD-deficient (G6PD-Med variant with <3-5% residual activity) indicate that the deficiency is perfectly well compensated (except for favism and kernicterus in infancy). Therefore, present reviewer considers the prospect of a pharmacological "therapy" and "correction" of G6PD-deficiency as unrealistic for the following reasons:

a) Life expectancy of G6PD-deficient subjects is significantly higher compared to normal subjects. For example, a cohort of G6PD-deficient men (Mediterranean variant) was followed during 11 years and compared with a normal G6PD cohort. At the trial end, the deficient cohort experienced a significantly reduced overall mortality mostly due to four-fold reduction of cerebrovascular and heart disease.

b) The prevalence of Sardinian G6PD-Med-deficient male centenarians was four-fold that of continental Italians (a, b: see AG Schwartz, LL Pashko, *Aging Res Rev.* 2004;3:171-187).

c) Clinical and mortality studies indicate that cardiovascular disease susceptibility (CDS) was significantly lower in G6PD-Med-deficient men (see Manganelli et al, *Cardiovasc Hematol Dis.* 2013;13:73-82).

R: The reviewer has noted cohort studies which demonstrate higher life expectancy and lower risk of cardiovascular disease in patients with G6PD deficiency. Whereas all these observations are noteworthy, they are correlative in nature and G6PD deficiency might not be the sole cause for these findings; compensatory mechanisms, such as upregulation of alternate enzymes that reduce cellular oxidative stress, may explain the phenotypes. There are also many other significant factors that could influence the phenotypes observed in the mentioned studies. These include more careful life style of the G6PD-deficient carriers, differences in diet, and other genetic traits that co-segregate with the G6PD mutant gene and provide evolutionary advantage to survive the deficiency.

Furthermore, contrary to the aforementioned studies, another study (see Hecker, PA. et al., *Circ Heart Fail.* 2013; 6(1):118-26) suggests that G6PD deficiency adversely affects the development of heart failure in response to cardiac stress, which highlights our limited knowledge on the role of G6PD deficiency in cardiovascular diseases. These authors also demonstrated in another publication (see Hecker, PA. et al., *Am J Physiol Heart Cir Physiol.* 2013; 304(4): H491-H500) that the protective effect of G6PD deficiency from cardiovascular disease development came from limited data. Thus, a more thorough evaluation in a large patient population is needed to

determine the effects of G6PD deficiency on the development of cardiovascular diseases and subsequent outcomes.

d) The risk of colorectal cancer was significantly reduced in G6PD-Med-deficient men (Dore et al, *Medicine*. 2016;95: 44) and suppression of G6PD-activity by DHEA (a powerful G6PD-inhibitor) showed inhibition of cervical cancer (Fang Z et al, *BBActa* 2016;1863:2245-2254). In more general terms though, contradictory results on the relationships between G6PD-deficiency and cancer are present in the literature (Cocco P, *J Epidemiol Community Health*. 1987;41(2):89-93).

R: The reviewer noted some beneficial effects of G6PD deficiency on cancer suppression. In the reference provided by the reviewer (Dore, MP. et al., *Medicine*. 2016;95: 44), the author provided some limitations of their findings that should have been taken into consideration such as unrepresentative populations in their study, lack of detailed data regarding other risk factors including diets, physical activity, and medications. As the reviewer also mentioned and cited the reference by Cocco, P., the correlation between G6PD and cancer is still ambiguous and controversial, which is supported by conflicting data (see Kuo, W. et al., *Int J Cancer*. 2000;85 (6): 857-64 [G6PD may act as a potential oncogene], Forteleoni, G. et al., *Tumori*. 1988; 74(6): 665-7 [G6PD Mediterranean allele does not play a protective role against the development of breast cancer], and Pisano, M. et al., *Tumori*. 1991; 77(1): 12-5 [G6PD deficiency does not provide significant protection against the development of lung cancer in humans]). Thus, further studies are needed to fully understand the hypothesis of the protective effect of G6PD deficiency against the development of cancers, including the pharmacological tool developed in our study.

e) By contrast with a)-d), published evidence supports the positive correlation between G6PD-deficiency and diabetes (see Meloni et al, *Haematologica*, 1992;77:94-95; Heymann et al, *Diabetes Care* 2012;35:e58), cataract (see Moro et al, *Am J Hum Genet* 1985;5:197-200) and cholelithiasis (see Meloni et al, *Acta Haematol* 1991;85:76-78).

R: In the revised manuscript, we have added references on positive correlation between G6PD deficiency and diabetes, cataract, and cholelithiasis (Discussion, p.13). We also corrected the sentence of “the anti-oxidant property of G6PD protects from a variety of other pathologies” to “the anti-oxidant property of G6PD may correlate to development of a variety of other pathologies” (p.13).

f) Contrary to the present Authors' suggestion to treat deficient subjects with G6PD-activity enhancers, the exact opposite has been suggested by others, namely to block G6PD-activity aimed at alleviating G6PD-deficiency-dependent pathologies. Rationale and prospects of this suggestion, based on administration of dehydroepiandrosterone (DHEA), a powerful inhibitor of G6PD, is summarized in a review article by Schwartz and Pashko (*Ageing Res Rev* 2004;3: 171-187).

R: In addition to acting as a G6PD inhibitor, DHEA is an adrenal steroid with many other molecular actions (some references below are cited as examples). Therefore, any conclusion about *in vivo* use of this G6PD inhibitor is difficult to interpret.

- Biaglow, JE. et al., *Biochem Biophys Res Commun*. 2000; 273(3): 846-52 [reporting that inhibition of cell growth is not G6PD related].
- Ng, HP. et al., *Food Chem Toxicol*. 1999; 37(3): 503-8 [reporting that DHEA may compensate for vitamin E deficiency].
- Paulin, R. et al., *Am J Physiol Heart Circ Physiol*. 2011; 301(5): H1798-809 [reporting that DHEA inhibits the Src/STAT3 in pulmonary arterial hypertension].
- Ho, HY. et al., *Int J Oncol*. 2008; 33(5): 969-77 [reporting that DHEA does not act *via* inhibition of G6PD and rather suppresses cell growth by altering mitochondrial gene expression, morphology, and function].

- Di Monaco, M. et al., *Br J Cancer*. 1997; 75(4): 589-92 [reporting that their study failed to confirm DHEA's putative anti-tumor action on breast cancer through G6PD inhibition].

g) Lastly, G6PD-deficiency provides distinct resistance against severe malaria. Making G6PD-deficient subjects pharmacologically normal would obliterate such essential mechanism of anti-malaria defense. Interestingly, the triad: falciparum malaria, G6PD-deficiency and fava bean consumption was present in almost all (except sub-Saharan Africa where fava beans were not cultivated) malarial regions. As discussed below, vicine and convicine of faba beans elicit hemolysis ("favism") by the same mechanism as chloroquine.

R: We agree that G6PD deficiency was probably selected for in human evolution, as a protective mechanism from malaria. The deficiency results in increased ROS levels in erythrocytes bearing the parasites and thus causing their death. Also, *Plasmodium* depends on obtaining nutrients from the erythrocytes, including ribose-5-phosphate and NADPH, which are lower in G6PD-deficient cells. The deficiency also increases membrane stiffness and decreases deformation of the erythrocyte membrane, thus reducing merozoite invasion of erythrocytes through endocytosis (see Zhang et al., *Biochem Biophys Acta* 2017;1864: 771-78). We added a sentence of "G6PD deficiency provides resistance against malaria" on p. 3.

Note also, that our study suggests another use of G6PD activator, for *ex vivo* preservation of blood for transfusion. Time-dependent biochemical and morphological changes of erythrocytes occur during storage process, which may contribute to some pathologies following transfusion such as increased risk of morbidity, longer hospitalizations, multiple organ failure, and a higher risk of mortality (see McKenny, M. et al., *Br J Anaesth* 2011; 106:643-649, Alexander, JT. et al., *Transfusion* 2013; 53:2619-2628, Spinella, PC. et al., *Transfusion* 2011; 51: 894-900, and Koch, CG. et al., *Ann Thorac Surg* 2013; 96:1894-99). Our work (Figure 5f-h) as well as a recent report show the progressive decline of G6PD activity of stored blood from non-G6PD-deficient donors. This decrease in G6PD activity has the potential to contribute to oxidative damage and deterioration of erythrocytes with storage (see Peters, AL. et al., *Transfusion*. 2016; 56(2):427-32). Thus, small molecules like AG1, which preserves G6PD activity, may prolong preservation of blood for transfusion (now discussed on p.12).

2. The Authors correctly emphasize the inability of the G6PD-deficient RBCs to keep glutathione in the reduced state upon oxidative stress, and more generally the propensity of deficient RBC to oxidative hemolysis (and kernicterus). In fact, those are the most frequent severe pathologies observed in G6PD-deficiency.

Specifically, G6PD-deficient subjects are at risk of severe hemolysis upon administration of primaquine, an essential drug for treatment of vivax malaria. Origin and implications of G6PD-deficiency and primaquine toxicity in vivax malaria are well described in K. Baird's review article in *Pathogens and Global Health*, 2015;109:93-106. The present authors underscore the importance of their suggested "therapy" of G6PD-deficiency, emphasizing the reversal (or abrogation) of primaquine-elicited hemolysis through increase in the activity of RBC G6PD.

Evidence and discussion of reversal of PQ hemolysis is though are not satisfactory for the following reasons:

a) Primaquine (PQ) hemolysis and favism, ie hemolysis after consumption of fava beans (FB) in G6PD-deficient subjects, are superimposable. Both are characterized by the fast, profound and irreversible oxidation of GSH in the RBC. GSH oxidation starts a chain of molecular events that finally lead to massive phagocytic elimination of RBC and anemia.

R: Thank you for your comments. We agree that we should have expanded on that and now added a brief discussion and this reference to our manuscript (Introduction, p.3).

b) While the molecular details of PQ hemolysis are scarce, favism hemolysis has been described in detail (see for example Luzzatto L, Arese P. Favism and glucose-6-phosphate dehydrogenase deficiency. *N Engl J Med.* 2018 Jan 4;378(1):60-71; Arese P, Pathophysiology of hemolysis in glucose-6-phosphate dehydrogenase deficiency. *Semin Hematol*, 1990;27:1-40).

R: Thank you for your comments. A brief discussion and the reference are now added to our manuscript (Introduction, p.3).

c) In summary, primaquine or vicine and convicine, the two redox-active compounds present in very high concentrations in FB, oxidize GSH in G6PD-deficient RBC. Due to insufficient NADPH production, the NADPH-dependent enzyme glutathione reductase (the GSH-regenerating enzyme) does not work. For this reason, reduced glutathione is not regenerated. Low GSH does not allow RBC protection against ROS radicals and hydrogen peroxide continuously generated in the RBC; ROS radicals and peroxide oxidize RBC protein thiols and lipids, generate the powerful oxidants ferryl Hb and hemichromes and cause iron release from Hb. RBC membrane proteins are oxidized and clustered, generating Heinz bodies. Finally, clustered band 3 leads to membrane deposition of autologous IgG and complement, leading to RBC phagocytosis and extravascular hemolysis (EH). EH is the vastly predominant form of RBC destruction in favism and primaquine hemolysis.

R: We thank the reviewer for the expert advice and suggestion and have now added new data and a brief discussion of the molecular mechanism of hemolysis (main text describing the new Figure 5, p. 12). When RBC isolated from whole blood samples (collected from several de-identified subjects) were treated with either chloroquine or diamide (1 mM), we found that band 3 protein is clustered and located at the top of the gel, as the reviewer predicted. Importantly, and consistent with our work on AG1-induced reduction of hemolysis, we further found that AG1 abrogated the clustering of band 3 protein. The Western blot data and their quantification are now provided in new Figure 5e and Supplementary Figure 7a.

d) Importantly, if deficient G6PD was indeed activated by AG1 treatment (as asserted by the Authors), we should expect a remarkable GSH regeneration in the isolated, deficient RBC pre-treated with a specific GSH oxidant, such as FB components or diamide or similar GSH oxidants. This essential proof of evidence for AG1-induced G6PD activation and GSH regeneration is sorely missing.

R: We agree with the reviewer and have added a new Figure 5, with data addressing this comment. In short, isolated RBCs from several subjects with normal G6PD activity (see Method section, p. 22) that were treated with either 1 mM chloroquine or diamide exhibited hemolysis, elevated ROS production and decreased GSH levels, which corresponded with decreased G6PD activity, and all were greatly blunted by AG1 treatment.

e) In conclusion, there is no evidence that AG1 works and re-activates the mutant G6PD.

R: The data provided in new Figure 5, as well as in Figure 4 d-f [AG1 did not affect any phenotypic or biochemical parameters in G6PD knockout zebrafish], Supplementary Figure 6c,d [pericardial edema developed due to mesoderm defects in *tbx16* mutants was not corrected by AG1 treatment], Supplementary Figure 3d [AG1 did not improve cell viability when G6PD was knocked down in SH-SY5Y cells by siRNA], and Supplementary Figure 5c [AG1 did not improve cell viability of G6PD-deficient fibroblasts (carrying a Mediterranean mutation in G6PD) when G6PD was knocked down by siRNA] all support direct evidence that AG1 acts *via* activation of G6PD.

Reviewer #3:

Overall this is a well-written concise manuscript with novel findings and possible translation into the clinical realm. There is very little work in the field of small molecule discovery for G6PD deficiency, making this work novel, important to publish, and likely high impact.

R: Thank you for your encouraging comments.

I have no concerns about the manuscript or experiments shown.

Comments:

Did the authors have access to Canton G6PD (or A-) blood to test AG1?

R: We have recently obtained only a few de-identified G6PD-deficient blood samples from the RBC Special Studies Laboratory at Stanford University Medical Center. [Due to the amount of blood samples provided from the lab,

we could not determine what their genotype is, nor perform extensive studies.] Based on their G6PD activity, the RBC lab and our lab both confirmed that the subjects have G6PD deficiency (Figure 3a; normal G6PD activity is in the range of 5-9 U/g Hb). We found that AG1 increased mutant G6PD activity by 43% (Figure 3b). Also, G6PD-deficient RBC hemolyzed much faster and greater than normal RBC over time (Figure 3c), which was partially reduced by AG1 treatment (Figure 3c, d).

With an IRB protocol approval, we are in the process of recruiting G6PD-deficient subjects. Our SAR effort will be focused on identifying for effective AG1 analogs.

Figure 3 - **a** G6PD activity measured in the eight independent G6PD-deficient subjects. **b** Normalized G6PD activity (n=8). **c** Hemolysis assay when G6PD-deficient RBC was stored at refrigerated temperature with AG1 for 28 days. **d** Time course of hemolysis for 28 days (n=3). Error bars represent mean \pm SD.

In the phenotyping of AG1 treated zebrafish, can authors show statistical improvement in the phenotypic (Chi-Square)?

R: We have now included the raw counts used for chi-square analysis and calculated p values in Figure 4a, 4d, Supplemental Figure 6b, and Supplemental Figure 6f.

How did the authors arrive at 1 µM concentration to be used in the zebrafish testing?

R: As provided in Supplemental Figure 6a and stated in the main text on p 10, we first tested the toxicity of AG1 on zebrafish development in the range of 1-10 µM. We found that 1-5 µM range is not toxic. Since this concentration range also corresponds to the concentration we used for cell-based assays and erythrocyte study, we used 1 µM also in the zebrafish study.

Was there a death phenotype in the zebrafish experiments and did AG1 rescue it?

R: We have not observed a death phenotype in the setting of the zebrafish experiments, which is consistent with the viability observed in a morpholino-based G6PD-deficient zebrafish model (see Patrinostró, X. et al., *Exp Hematol.* 2013; 41:697-710).

Reviewer #3:

Remarks to the Author:

I have reviewed the authors' edits and additions to the paper and am satisfied with their response.

Thank-you

REVIEWERS' COMMENTS: Reviewer #3 (Remarks to the Author): I have reviewed the authors' edits and additions to the paper and am satisfied with their response. Thank-you R: Thank you.